# Unlearning-Aware Minimization

**Hoki Kim**[*][†]
Chung-Ang University
hokikim@cau.ac.kr

**Keonwoo Kim**[†]
NAVER Digital Healthcare LAB
keonwoo.kim97@navercorp.com

**Sungwon Chae**
Seoul National University
csw0815@snu.ac.kr

**Sangwon Yoon**
Ministry of Justice, Republic of Korea
asd01075272750@gmail.com

## Abstract

Machine unlearning aims to remove the influence of specific training samples (i.e., forget data) from a trained model while preserving its performance on the remaining samples (i.e., retain data). Existing approximate unlearning approaches, such as fine-tuning or negative gradient, often suffer from either insufficient forgetting or significant degradation on retain data. In this paper, we introduce Unlearning-Aware Minimization (UAM), a novel min–max optimization framework for machine unlearning. UAM perturbs model parameters to maximize the forget loss and then leverages the corresponding gradients to minimize the retain loss. We derive an efficient optimization method for this min-max problem, which enables effective removal of forget data and uncovers better optima that conventional methods fail to reach. Extensive experiments demonstrate that UAM outperforms existing methods across diverse benchmarks, including image classification datasets (CIFAR-10, CIFAR-100, TinyImageNet) and multiple-choice question-answering benchmarks for large language models (WMDP-Bio, WMDP-Cyber).

## 1   Introduction

The increasing deployment of artificial intelligence (AI) into real-world applications has prompted critical discussions regarding the alignment of AI with human values. A key aspect of these discussions is the *"right to be forgotten"* [25], a principle embedded in the General Data Protection Regulation (GDPR) [14]. This right enables individuals to request the deletion of their personal data, providing a safeguard for privacy and mitigating risks associated with data misuse.

The simplest approach is retraining a model from scratch without the data points to be forgotten. However, retraining is computationally prohibitive, particularly for large-scale deep learning models [4, 35]. As a potential solution, machine unlearning has emerged for removing the influence of specific data samples by appropriately updating their parameters [2, 3, 9]. The goal of machine unlearning is to efficiently remove the influence of specific data while preserving performance on the remaining data. Formally, let $\mathcal{D}_f$ denote the dataset to be forgotten (i.e., *forget data*) and $\mathcal{D}_r$ denote the dataset to be retained (i.e., *retain data*). A common strategy of machine unlearning is to optimize the following objectives: minimizing the retain loss $\mathcal{L}(\boldsymbol{w}, \mathcal{D}_r)$ or maximizing the forget loss $\mathcal{L}(\boldsymbol{w}, \mathcal{D}_f)$ [11, 32].

In this paper, we introduce **Unlearning-Aware Minimization (UAM)**, a novel min-max optimization framework for machine unlearning. Specifically, UAM formulates unlearning as a two-stage process:

---

[*]Corresponding author: hokikim@cau.ac.kr

[†]Equal contribution

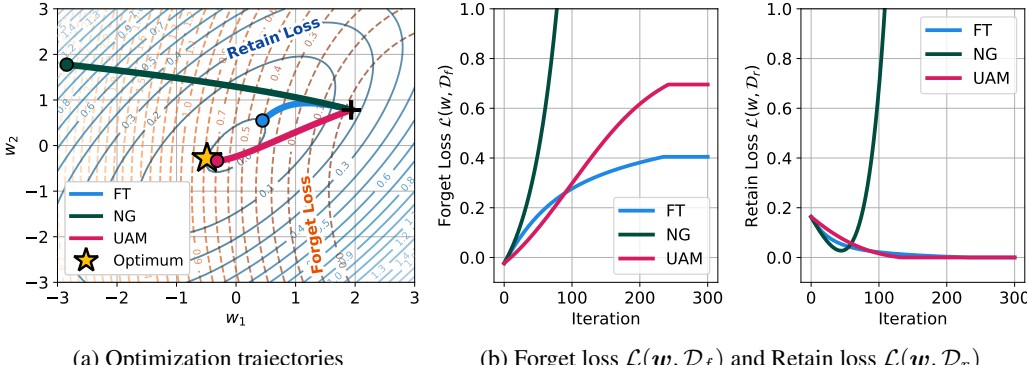

(a) Optimization trajectories

(b) Forget loss $\mathcal{L}(\boldsymbol{w}, \mathcal{D}_f)$ and Retain loss $\mathcal{L}(\boldsymbol{w}, \mathcal{D}_r)$

Figure 1: **Optimization results on synthetic loss functions.** The optimization begins at the global minimum of the sum of two losses (black cross, **+**). (a) UAM successfully converges to the optimal point (yellow star, ★), whereas other methods converge to suboptimal regions. For NG, only a partial trajectory is shown due to divergence. (b) UAM achieves a high forget loss while effectively minimizing the retain loss. NG yields a high forget loss but fails to maintain low retain loss. FT converges to a suboptimal region, resulting in a lower forget loss.

(i) an inner maximization that identifies a surrogate weight within a local neighborhood that maximizes the forget loss, and (ii) an outer minimization that reduces the retain loss by using its gradients. Intuitively, this procedure ensures that the model exhibits characteristics similar to those of weights with a high forget loss, while maintaining a low retain loss. By leveraging a first-order Taylor approximation, we derive a scalable algorithm that enables effective unlearning while remaining computationally practical.

Fig. 1 highlights the key differences between UAM and existing unlearning methods. Fine-tuning (FT) minimizes $\mathcal{L}(\boldsymbol{w}, \mathcal{D}_r)$ and negative gradient (NG) maximizes $\mathcal{L}(\boldsymbol{w}, \mathcal{D}_f)$ [11, 32]. Both methods converge to suboptimal regions in the given optimization problem. These methods result in either low forget loss (i.e., insufficient forgetting) or high retain loss (i.e., poor performance). In contrast, UAM more effectively navigates the loss landscape, approaching the optimal solution characterized by high forget loss and low retain loss. As shown in Fig. 1b, UAM achieves a higher forget loss and a lower retain loss by explicitly exploring regions with high forget loss. In our experiments, UAM demonstrates superior performance on both image classification and multiple-choice question answering tasks.

Our main contributions can be summarized as follows:

- We propose a new min-max optimization framework for machine unlearning, Unlearning-Aware Minimization (UAM). By leveraging model parameters with high forget loss, UAM enables the effective removal of forget data while preserving the performance on retain data.

- We establish an efficient algorithm based on a first-order Taylor expansion. We also provide a theoretical analysis of UAM, characterizing its optimization dynamics through the cosine similarity between retain and forget gradients.

- We evaluate the effectiveness of UAM on image classification datasets (CIFAR-10, CIFAR-100, and TinyImageNet) and multiple-choice question-answering datasets (WMDP-Bio and WMDP-Cyber) using large language model (LLM). Since UAM is a framework independent of any specific loss function, it can be easily extended to other domains.

- To promote reproducibility and benchmarking within the machine unlearning community, we release implementations of existing baseline unlearning methods, along with our proposed framework, available at: https://github.com/Harry24k/machine-unlearning-pytorch.

## 2 Related Work

### 2.1 Machine Unlearning

Machine unlearning [3, 9] aims to eliminate the influence of forget data $\mathcal{D}_f$, while preserving knowledge learned from retain data $\mathcal{D}_r$. The ideal solution, known as *exact unlearning*, is to retrain the model from scratch without $\mathcal{D}_f$; however, retraining is often computationally inefficient for large-scale deep learning models [4, 35]. Therefore, *approximate unlearning* methods have been developed. Fine-tuning (FT) [9, 32] minimizes $\mathcal{L}(\boldsymbol{w}, \mathcal{D}_r)$ relying catastrophic forgetting [24]; negative gradient (NG) [9], which also referred to as gradient ascent, directly maximizes $\mathcal{L}(\boldsymbol{w}, \mathcal{D}_f)$. More recent methods build on these frameworks by incorporating additional techniques such as distillation [20] or pruning [16]. In contrast to prior approaches, we highlight the potential of a min–max optimization framework for machine unlearning that extends beyond traditional dual-objective formulations. We note that there exists a distinct line of research, including Fisher Forgetting (FF) [9] and Influence Unlearning (IU) [15, 18], that leverages the Fisher Information Matrix and influence functions.

### 2.2 Min-Max Optimization

Min-max optimization refers to a class of learning problems that aims to solve two competing objectives, commonly formulated as saddle-point or bi-level optimization problems [1, 31]. In deep learning, the min-max optimization plays a central role in several research areas. For example, in adversarial robustness [10, 28], adversarial training uses an inner maximization to find perturbations that maximize a loss value, and an outer minimization to optimize model parameters to minimize this worst-case loss [23, 36]. More recently, sharpness-aware minimization has introduced a min-max optimization for improving generalization. It adopts an inner maximization and an outer minimization step to identify parameters that have uniformly low losses within neighborhoods [7, 21]. We extend a min-max optimization into the domain of machine unlearning using two disjoint datasets $\mathcal{D}_r$ and $\mathcal{D}_f$ and demonstrate that a min-max optimization can address the challenges in machine unlearning.

## 3 Unlearning-Aware Minimization

In this work, we use the following notation: scalars are denoted by $a$, vectors by $\boldsymbol{a}$, matrices by $\mathbf{A}$, and $\triangleq$ indicates equality by definition. Let us denote the training dataset as $\mathcal{D} \triangleq \{(\boldsymbol{x}_i, \boldsymbol{y}_i)\}_{i=1}^n$, drawn independently and identically (i.i.d.) from the true data distribution. A model parameterized by weights $\boldsymbol{w} \in \mathcal{W} \subseteq \mathbb{R}^d$ is trained by minimizing the empirical training loss $\mathcal{L}(\boldsymbol{w}, \mathcal{D}) \triangleq \frac{1}{n} \sum_{i=1}^n \ell(\boldsymbol{w}, x_i, y_i)$, where $\ell$ is an individual loss function. We denote the subset of data to be forgotten as the *forget data* $\mathcal{D}_f \subset \mathcal{D}$. Then, the complement of $\mathcal{D}_f$ becomes the *retain data* $\mathcal{D}_r \triangleq \mathcal{D} \setminus \mathcal{D}_f$.

The simplest and exact approach to unlearning, commonly called *exact unlearning*, optimizes a model from scratch using only the retain data:

$$\boldsymbol{w}^* = \arg\min_{\boldsymbol{w}} \mathcal{L}(\boldsymbol{w}, \mathcal{D}_r), \tag{1}$$

commonly known as *Retrain* [16] or the oracle [4]. While exact unlearning provides an optimal solution for eliminating the influence of the forget data, its substantial computational overhead makes it impractical for large-scale models and datasets [16, 29]. To circumvent these practical constraints, *approximate unlearning* methods often re-optimize the pre-trained model with $\boldsymbol{w}_0 = \arg\min_{\boldsymbol{w}} \mathcal{L}(\boldsymbol{w}, \mathcal{D})$. Therefore, approximate unlearning methods commonly assume that the solution of (1) lies within a bounded neighborhood $\mathcal{B}_\Omega(\boldsymbol{w}_0) \triangleq \{\tilde{\boldsymbol{w}} \in \mathbb{R}^d \mid \|\boldsymbol{w}_0 - \tilde{\boldsymbol{w}}\| \leq \Omega\}$, where $\Omega$ is a finite upper bound. Given that $\boldsymbol{w}^* \in \mathcal{B}_\Omega(\boldsymbol{w}_0)$, we can characterize existing approximate unlearning methods as approaches that aim to solve the following optimization problem, initialized at $\boldsymbol{w} = \boldsymbol{w}_0$:

$$\min_{\boldsymbol{w}} \mathcal{L}(\boldsymbol{w}, \mathcal{D}_r) + \beta \big[ \mathcal{L}(\boldsymbol{w}^*, \mathcal{D}) - \mathcal{L}(\boldsymbol{w}, \mathcal{D}) \big], \tag{2}$$

where $\beta$ is a hyperparameter for balancing two different losses. The first term, $\mathcal{L}(\boldsymbol{w}, \mathcal{D}_r)$, encourages the model to maintain performance on the retain data $\mathcal{D}_r$. The second term encourages alignment (or consistency) between the optimized weights $\boldsymbol{w}$ and the solution $\boldsymbol{w}^*$. Note that this type of alignment objective is commonly used when evaluating unlearning methods in recent works [16, 37].

The objective in equation (2) offers a unified explanation for two key approximate unlearning methods: FT and NG. First, setting $\beta = 0$ simplifies the objective to the objective of FT, $\min_{\boldsymbol{w}} \mathcal{L}(\boldsymbol{w}, \mathcal{D}_r)$.

Since FT ignores the second term with $\boldsymbol{w}^*$ in (2), it results in poor forgetting performance as shown in Fig. 1. For NG, we establish the following lemma:

**Lemma 1.** *For $\beta = |\mathcal{D}|/|\mathcal{D}_r|$, which balances the two loss terms based on the number of data points, the objective (2) becomes*

$$\min_{\boldsymbol{w}} \mathcal{L}(\boldsymbol{w}^*, \mathcal{D}_r) + \frac{|\mathcal{D}_f|}{|\mathcal{D}_r|} \big[ \mathcal{L}(\boldsymbol{w}^*, \mathcal{D}_f) - \mathcal{L}(\boldsymbol{w}, \mathcal{D}_f) \big]. \tag{3}$$

*Proof.* See Appendix. $\square$

In (3), assuming that there is no prior knowledge of $\boldsymbol{w}^*$ (i.e., $d\boldsymbol{w}^*/d\boldsymbol{w} = 0$), the objective is reduced to NG, which depends solely on the gradient $\nabla_{\boldsymbol{w}} \mathcal{L}(\boldsymbol{w}, \mathcal{D}_f)$. However, since NG focuses solely on $\max_{\boldsymbol{w}} \mathcal{L}(\boldsymbol{w}, \mathcal{D}_f)$, it struggles to maintain accuracy on $\mathcal{D}_r$.

Rather than ignoring $\boldsymbol{w}^*$, we propose to use a surrogate weight $\hat{\boldsymbol{w}}$ that characterizes $\boldsymbol{w}^*$. A key insight is that $\boldsymbol{w}^*$ lies within a neighborhood where the forget loss $\mathcal{L}(\boldsymbol{w}, \mathcal{D}_f)$ remains sufficiently high. To formalize this, we introduce the following definition:

**Definition 1.** ($\epsilon$-forget neighborhood) Given parameters $\boldsymbol{w}$, forget data $\mathcal{D}_f$, and threshold $\epsilon > 0$, the $\epsilon$-forget neighborhood is defined as:

$$\mathcal{B}_{\Omega}^{\epsilon}(\boldsymbol{w}; \mathcal{D}_f) \coloneqq \big\{ \tilde{\boldsymbol{w}} \in \mathbb{R}^d \mid \mathcal{L}(\tilde{\boldsymbol{w}}, \mathcal{D}_f) \geq \epsilon, \ \|\boldsymbol{w} - \tilde{\boldsymbol{w}}\| \leq \Omega \big\}. \tag{4}$$

This set characterizes the region where a weight has a forget loss of at least $\epsilon$, ensuring that the influence of the forget data is sufficiently removed. Therefore, for an appropriately chosen $\epsilon$, we have $\boldsymbol{w}^* \in \mathcal{B}_{\Omega}^{\epsilon}(\boldsymbol{w}_0; \mathcal{D}_f)$. Since the exact $\boldsymbol{w}^*$ is intractable, we introduce a surrogate weight $\hat{\boldsymbol{w}}$ to characterize the high-forget-loss characteristic as follows:

$$\hat{\boldsymbol{w}} \triangleq \arg\max_{\|\boldsymbol{\delta}\|_2 \leq \rho} \mathcal{L}(\boldsymbol{w} + \boldsymbol{\delta}, \mathcal{D}_f), \tag{5}$$

where $\rho$ is a radius that satisfies $\hat{\boldsymbol{w}} \in \mathcal{B}_{\Omega}^{\epsilon}(\boldsymbol{w}_0; \mathcal{D}_f)$. While this surrogate weight $\hat{\boldsymbol{w}}$ becomes dynamic in contrast to the fixed optimal weight $\boldsymbol{w}^*$, it provides a practical way of approximating the behavior of $\boldsymbol{w}^* \in \mathcal{B}_{\Omega}^{\epsilon}(\boldsymbol{w}_0; \mathcal{D}_f)$. Substituting this surrogate weight into (3) reformulates the problem as the following min–max optimization:

$$\min_{\boldsymbol{w}} \mathcal{L}(\arg\max_{\|\boldsymbol{\delta}\|_2 \leq \rho} \mathcal{L}(\boldsymbol{w} + \boldsymbol{\delta}, \mathcal{D}_f), \mathcal{D}_r) + \frac{|\mathcal{D}_f|}{|\mathcal{D}_r|} \big[ \mathcal{L}(\arg\max_{\|\boldsymbol{\delta}\|_2 \leq \rho} \mathcal{L}(\boldsymbol{w} + \boldsymbol{\delta}, \mathcal{D}_f), \mathcal{D}_f) - \mathcal{L}(\boldsymbol{w}, \mathcal{D}_f) \big]. \tag{6}$$

This min-max optimization can be simplified into an efficient algorithm by approximating the inner maximization problem under the first-order approximation.

**Theorem 1.** *(Efficient min-max optimization for approximate unlearning) Suppose that $\mathcal{L}(\boldsymbol{w}, \mathcal{D}_f)$ can be locally approximated by its first-order Taylor expansion around $\boldsymbol{w}$. Then, the min-max optimization objective in (6) can be simplified to:*

$$\min_{\boldsymbol{w}} \mathcal{L}(\boldsymbol{w} + \rho \frac{\nabla_{\boldsymbol{w}} \mathcal{L}(\boldsymbol{w}, \mathcal{D}_f)}{||\nabla_{\boldsymbol{w}} \mathcal{L}(\boldsymbol{w}, \mathcal{D}_f)||_2^2}, \mathcal{D}_r). \tag{7}$$

*Proof.* Applying a first-order Taylor expansion, the inner maximization of (6) can be approximated to

$$\max_{||\boldsymbol{\delta}||_2 \leq \rho} \mathcal{L}(\boldsymbol{w} + \boldsymbol{\delta}, \mathcal{D}_f) \approx \max_{||\boldsymbol{\delta}||_2 \leq \rho} \mathcal{L}(\boldsymbol{w}, \mathcal{D}_f) + \boldsymbol{\delta}^T \nabla_{\boldsymbol{w}} \mathcal{L}(\boldsymbol{w}, \mathcal{D}_f). \tag{8}$$

Let us denote $\mathcal{L}_f(\boldsymbol{w}) \triangleq \mathcal{L}(\boldsymbol{w}, \mathcal{D}_f)$. The solution to this optimization problem, known as standard dual-norm arguments [7, 28], is given explicitly by $\boldsymbol{\delta} = \rho \nabla_{\boldsymbol{w}} \mathcal{L}_f(\boldsymbol{w}) / ||\nabla_{\boldsymbol{w}} \mathcal{L}_f(\boldsymbol{w})||_2^2$. Unless specified otherwise, $\nabla = \nabla_{\boldsymbol{w}}$. Substituting the solution $\boldsymbol{\delta}$ into the second term on the right-hand side of (6), we have

$$\mathcal{L}_f(\boldsymbol{w} + \rho \frac{\nabla \mathcal{L}_f(\boldsymbol{w})}{||\nabla \mathcal{L}_f(\boldsymbol{w})||_2^2}) - \mathcal{L}_f(\boldsymbol{w}) \approx \big[ \mathcal{L}_f(\boldsymbol{w}) + \big( \rho \frac{\nabla \mathcal{L}_f(\boldsymbol{w})}{||\nabla \mathcal{L}_f(\boldsymbol{w})||_2^2} \big)^T \nabla \mathcal{L}_f(\boldsymbol{w}) \big] - \mathcal{L}_f(\boldsymbol{w}) \tag{9}$$

$$\approx \big[ \mathcal{L}_f(\boldsymbol{w}) + \rho \big] - \mathcal{L}_f(\boldsymbol{w}) = \rho. \tag{10}$$

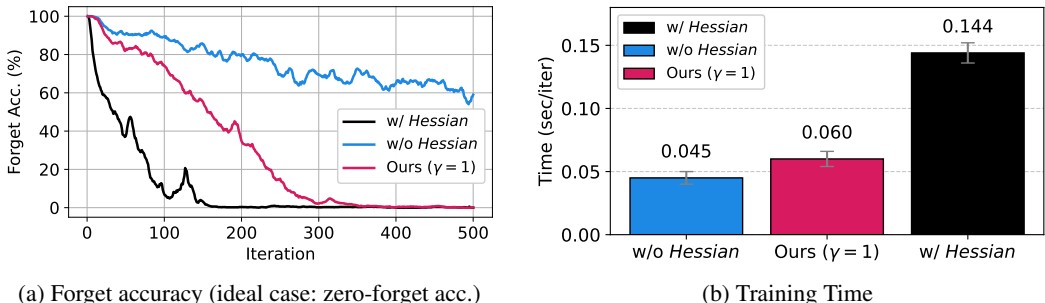

(a) Forget accuracy (ideal case: zero-forget acc.)    (b) Training Time

Figure 2: **Effectiveness of** $\gamma$. (CIFAR-10, Class-wise forgetting) We compare three optimizations: *w/ Hessian* corresponds to the exact computation of (12); *w/o Hessian* corresponds to the omission of the second-order gradient term in (13); Ours indicates the relaxed optimization in (15). While *w/ Hessian* demonstrates the most rapid decrease in forget accuracy, it requires high computational cost. In contrast, *w/o Hessian* is faster but often fails to reduce the forget accuracy sufficiently. Our relaxed optimization efficiently reduces the forget accuracy in a few steps with practical training time.

Hence, the optimization (6) can be approximated to

$$\min_{\boldsymbol{w}} \mathcal{L}\left(\boldsymbol{w} + \rho \frac{\nabla \mathcal{L}(\boldsymbol{w}, \mathcal{D}_f)}{||\nabla \mathcal{L}(\boldsymbol{w}, \mathcal{D}_f)||_2^2}, \mathcal{D}_r\right) + \frac{|\mathcal{D}_f|}{|\mathcal{D}_r|}\rho. \tag{11}$$

Since the second term $\frac{|\mathcal{D}_f|}{|\mathcal{D}_r|}\rho$ does not depend on $\boldsymbol{w}$, we have the simplified optimization objective stated in the theorem. □

Theorem 1 allows us to directly apply stochastic gradient descent. The gradient of the objective function (7) can be explicitly computed using the following lemma.

**Lemma 2.** *(Update gradient of (7)) The update gradient of the objective function in (7) is given by:*

$$\nabla_{\boldsymbol{w}} \mathcal{L}(\boldsymbol{w} + \delta(\boldsymbol{w}), \mathcal{D}_r) = \left[\mathbf{I} + \frac{\rho}{||\nabla_{\boldsymbol{w}} \mathcal{L}(\boldsymbol{w}, \mathcal{D}_f)||_2^2}(\mathbf{I} - 2\mathbf{P}_f)\mathbf{H}_f\right] \nabla_{\boldsymbol{w}} \mathcal{L}(\boldsymbol{w}, \mathcal{D}_r)|_{\boldsymbol{w}+\delta(\boldsymbol{w})}, \tag{12}$$

*where $\mathbf{P}_f$ is the orthogonal projection matrix onto the space spanned by $\nabla_{\boldsymbol{w}} \mathcal{L}(\boldsymbol{w}, \mathcal{D}_f)$, and $\mathbf{H}_f$ denotes the Hessian of $\mathcal{L}(\boldsymbol{w}, \mathcal{D}_f)$.*

*Proof.* By chain rule,

$$\nabla_{\boldsymbol{w}} \mathcal{L}(\boldsymbol{w} + \delta(\boldsymbol{w}), \mathcal{D}_r) = \nabla_{\boldsymbol{w}} \mathcal{L}(\boldsymbol{w}, \mathcal{D}_r)|_{\boldsymbol{w}+\delta(\boldsymbol{w})} + \frac{d\delta(\boldsymbol{w})}{d\boldsymbol{w}} \nabla_{\boldsymbol{w}} \mathcal{L}(\boldsymbol{w}, \mathcal{D}_r)|_{\boldsymbol{w}+\delta(\boldsymbol{w})}. \tag{13}$$

Let $g(\boldsymbol{w}) \triangleq \nabla_{\boldsymbol{w}} \mathcal{L}(\boldsymbol{w}, \mathcal{D}_f)$ and $r(\boldsymbol{w}) \triangleq ||g(\boldsymbol{w})||_2^2$. Then, $\delta(\boldsymbol{w}) = \rho g(\boldsymbol{w})/r(\boldsymbol{w})$. Since $\frac{dg(\boldsymbol{w})}{d\boldsymbol{w}} = \mathbf{H}_f$ and $\frac{dr(\boldsymbol{w})}{d\boldsymbol{w}} = 2g(\boldsymbol{w})^T \mathbf{H}_f$, we have

$$\frac{d\delta(\boldsymbol{w})}{d\boldsymbol{w}} = \frac{\rho}{r(\boldsymbol{w})} \left(\mathbf{I} - 2\frac{g(\boldsymbol{w})g(\boldsymbol{w})^T}{r(\boldsymbol{w})}\right) \mathbf{H}_f. \tag{14}$$

□

In (12), the computation of the Hessian matrix is computationally expensive [7, 15, 18]. Therefore, some might suggest that omitting the second term $\frac{d\delta(\boldsymbol{w})}{d\boldsymbol{w}} \nabla_{\boldsymbol{w}} \mathcal{L}(\boldsymbol{w}, \mathcal{D}_r)|_{\boldsymbol{w}+\delta(\boldsymbol{w})}$ in (13), which is a technique used in several prior works across different domains [7, 10]. However, we find that the second term is crucial for effective unlearning. In Fig. 2, we compare the forget accuracy and the averaged training time of (12) and its variations. Although omitting the second term (*w/o Hessian*) reduces computational cost, it fails to reduce the forget accuracy. On the other hand, computing the exact (12) (*w/ Hessian*) achieves a lower forget accuracy, but it requires nearly three times the computational cost.

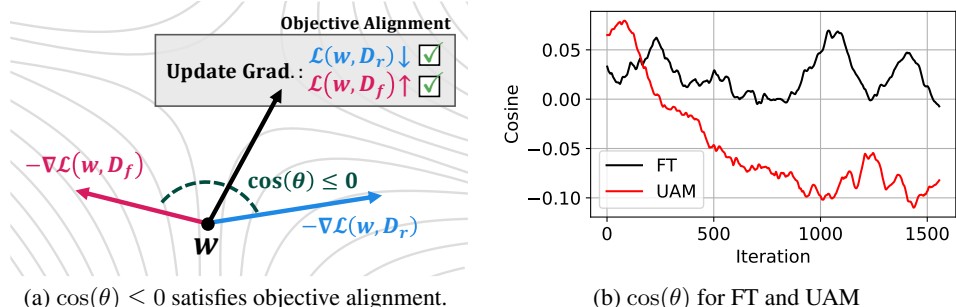

(a) $\cos(\theta) \leq 0$ satisfies objective alignment.    (b) $\cos(\theta)$ for FT and UAM

Figure 3: **Geometric interpretation of UAM**. (CIFAR-10, Class-wise forgetting) Compared to FT, UAM explicitly minimizes the cosine similarity, $\cos(\theta)$, between the retain gradient and the forget gradient. (a) When $\cos(\theta) \leq 0$, reducing the retain loss inherently leads to an increase in the forget loss. (b) The moving average of $\cos(\theta)$ in UAM shows a clear decreasing trend, with negative values.

Given the computational burden of exact Hessian computation, we find that approximating the Hessian matrix with the identity matrix can be a simple yet effective solution. This yields an efficient gradient update by introducing a hyperparameter $\gamma$:

$$[\mathbf{I} - \gamma \mathbf{P}_f] \nabla_{\boldsymbol{w}} \mathcal{L}(\boldsymbol{w}, \mathcal{D}_r)|_{\boldsymbol{w}+\delta(\boldsymbol{w})}. \tag{15}$$

As shown in Fig. 2, our relaxed optimization effectively reduces the forget accuracy with a small increase in computational time since the forget gradient $\nabla_{\boldsymbol{w}} \mathcal{L}(\boldsymbol{w}, \mathcal{D}_f)$ is already computed during the inner maximization. Our analysis with the projection matrix $\mathbf{P}_f$ offers a theoretical explanation for recent work [17], which utilizes the projection of retain and forget gradients to achieve unlearning in image generation tasks. For a more detailed discussion, including an ablation study on $\gamma$, please refer to Appendix. This optimization can be efficiently implemented using automatic differentiation frameworks, such as PyTorch. We name this framework **Unlearning-Aware Minimization (UAM)**.

**Geometric interpretation of UAM.**  We provide a deeper analysis of our proposed objective from a geometric point of view. By applying a first-order Taylor expansion around $\boldsymbol{w}$ on (7), we have:

$$\min_{\boldsymbol{w}} \mathcal{L}\left(\boldsymbol{w} + \rho \frac{\nabla \mathcal{L}(\boldsymbol{w}, \mathcal{D}_f)}{||\nabla \mathcal{L}(\boldsymbol{w}, \mathcal{D}_f)||_2^2}, \mathcal{D}_r\right) \approx \min_{\boldsymbol{w}} \mathcal{L}(\boldsymbol{w}, \mathcal{D}_r) + \nabla \mathcal{L}(\boldsymbol{w}, \mathcal{D}_r)^\top \rho \frac{\nabla \mathcal{L}(\boldsymbol{w}, \mathcal{D}_f)}{||\nabla \mathcal{L}(\boldsymbol{w}, \mathcal{D}_f)||_2^2}. \tag{16}$$

Compared to FT, which only optimizes the first term $\min_{\boldsymbol{w}} \mathcal{L}(\boldsymbol{w}, \mathcal{D}_r)$ in (16), UAM explicitly minimizes the inner product between $\nabla \mathcal{L}(\boldsymbol{w}, \mathcal{D}_r)$ and $\nabla \mathcal{L}(\boldsymbol{w}, \mathcal{D}_f)$, which is proportional to their cosine similarity. This implicit objective encourages the gradients $\nabla \mathcal{L}(\boldsymbol{w}, \mathcal{D}_r)$ and $\nabla \mathcal{L}(\boldsymbol{w}, \mathcal{D}_f)$ to become less aligned, or even negatively aligned. From a geometric perspective, as illustrated in Fig. 3a, a negative cosine similarity between these gradients indicates that minimizing the retain loss inherently removes learned information from the forget data.

Fig. 3b shows the historical value of $\cos(\theta)$ where $\theta$ is the angle between $\nabla \mathcal{L}(\boldsymbol{w}, \mathcal{D}_r)$ and $\nabla \mathcal{L}(\boldsymbol{w}, \mathcal{D}_f)$ during training. To reduce the stochasticity introduced by batch training, we use the first batch to measure $\cos(\theta)$. UAM exhibits a clear decreasing trend, resulting in a negative $\cos(\theta)$ compared to FT. Given the high dimensionality of the parameter space, this negative $\cos(\theta)$ provides a potential geometric explanation for the strong performance of UAM.

These results also align with the result shown in Fig. 1. We design a synthetic 2D optimization landscape with $\boldsymbol{w} = [w_1, w_2]$, where the forget and retain losses are derived from simple rotated quadratic forms as follows (more details in Appendix):

$$\mathcal{L}_f(\boldsymbol{w}) = \frac{1}{50}\left[3(w_1 - 2)^2 + (w_2 + 4)^2 - 6w_2 - 10w_1\right]; \mathcal{L}_r(\boldsymbol{w}) = \max\left(\frac{5r_\theta(w_1)^2 + r_\theta(w_2)^2}{50}, \delta\right),$$

$$\tag{17}$$

The optimization is initialized from the point that minimizes the sum of forget loss $\mathcal{L}_f(\boldsymbol{w})$ and retain loss $\mathcal{L}_r(\boldsymbol{w})$, representing a pre-trained model. Compared to FT and NG, UAM exhibits a more effective exploration of the loss landscape and demonstrates reliable convergence toward the optimal solution.

Table 1: **Machine unlearning performance on CIFAR-10.** The values in **blue** indicate the absolute differences from Retrain. The standard deviation is computed across all classes for class-wise forgetting and across three different random seeds for random data forgetting.

| Method | RA | FA | TA | $\Delta$Acc.($\downarrow$) | MIA-Eff. | Time |
|---|---|---|---|---|---|---|
| **Class-wise forgetting** | | | | | | |
| Retrain | $100.00_{\pm0.00}$ | $0.00_{\pm0.00}$ | $95.32_{\pm0.52}$ | 0.00 | $100.00_{\pm0.00}$ | 31.34 |
| FT | $100.00_{\pm0.00}$ (0.00) | $43.31_{\pm7.31}$ (43.31) | $95.01_{\pm0.55}$ (0.32) | 43.63 | $100.00_{\pm0.01}$ (0.00) | 1.56 |
| NG | $88.57_{\pm5.68}$ (11.43) | $0.80_{\pm1.40}$ (0.80) | $83.09_{\pm4.98}$ (12.23) | 24.45 | $99.34_{\pm1.15}$ (0.66) | 1.57 |
| FF | $99.98_{\pm0.05}$ (0.02) | $9.96_{\pm7.30}$ (9.96) | $95.18_{\pm0.66}$ (0.24) | 10.22 | $100.00_{\pm0.00}$ (0.00) | 0.97 |
| IU | $94.24_{\pm4.13}$ (5.76) | $0.27_{\pm0.48}$ (0.27) | $88.02_{\pm3.95}$ (7.29) | 13.32 | $99.99_{\pm0.02}$ (0.01) | 0.52 |
| $\ell_1$-sparse | $100.00_{\pm0.00}$ (0.00) | $0.00_{\pm0.00}$ (0.00) | $91.87_{\pm0.84}$ (3.45) | 3.45 | $100.00_{\pm0.00}$ (0.00) | 1.66 |
| UAM | $100.0_{\pm0.00}$ (0.01) | $0.00_{\pm0.00}$ (0.00) | $94.51_{\pm0.63}$ (0.81) | **0.82** | $100.00_{\pm0.00}$ (0.00) | 2.56 |
| **Random data forgetting** | | | | | | |
| Retrain | $100.00_{\pm0.00}$ | $95.33_{\pm0.39}$ | $94.73_{\pm0.14}$ | 0.00 | $11.86_{\pm0.34}$ | 31.32 |
| FT | $100.00_{\pm0.00}$ (0.00) | $99.66_{\pm0.09}$ (4.33) | $94.58_{\pm0.04}$ (0.15) | 4.48 | $4.43_{\pm0.36}$ (7.43) | 1.56 |
| NG | $61.97_{\pm4.29}$ (38.03) | $53.50_{\pm2.83}$ (41.83) | $58.83_{\pm4.04}$ (35.90) | 115.77 | $46.55_{\pm2.69}$ (34.69) | 1.56 |
| FF | $65.78_{\pm48.19}$ (34.22) | $65.49_{\pm48.31}$ (29.85) | $62.37_{\pm45.15}$ (32.36) | 96.43 | $10.30_{\pm0.50}$ (1.56) | 0.96 |
| IU | $97.30_{\pm2.44}$ (2.70) | $97.05_{\pm2.48}$ (2.43) | $91.06_{\pm2.68}$ (3.67) | 8.81 | $5.26_{\pm3.58}$ (6.60) | 0.53 |
| $\ell_1$-sparse | $100.00_{\pm0.00}$ (0.00) | $99.52_{\pm0.13}$ (4.19) | $94.48_{\pm0.18}$ (0.25) | 4.44 | $7.14_{\pm0.36}$ (4.72) | 1.66 |
| UAM | $99.88_{\pm0.03}$ (0.12) | $95.19_{\pm0.28}$ (0.41) | $92.74_{\pm0.29}$ (2.00) | **2.53** | $9.16_{\pm0.14}$ (2.70) | 2.55 |

# 4 Experiments

In this section, we conduct experiments on two major machine unlearning tasks: *image classification* and *multiple-choice question-answering* with large language model (LLM).

## 4.1 Image Classification

**Setup and Methods.** We conduct image classification experiments using three datasets: CIFAR-10, CIFAR-100 [19], and TinyImageNet [5]. We adopt ResNet-18 [12] for the CIFAR datasets and VGG [26] for TinyImageNet. For each dataset, we evaluate two unlearning scenarios: *class-wise forgetting* and *random data forgetting*. In the class-wise forgetting scenario, the forget set $\mathcal{D}_f$ consists of all training samples from a single class. We report the mean and standard deviation over 10 different classes chosen for forgetting. In the random data forgetting scenario, $\mathcal{D}_f$ consists of randomly sampled training examples across all classes. Results are averaged over three different random seeds.

We explore four representative unlearning frameworks: Fine-tuning (**FT**) [9, 32], Negative Gradient (**NG**) [9], Fisher Forgetting (**FF**) [9], Influence Unlearning (**IU**), [15, 18], and $\ell_1$-**sparse** [16]. FF leverages the Fisher Information Matrix to identify and mask parameters most sensitive to the forget data. IU uses the influence of each data point on the parameters. $\ell_1$-sparse encourages parameter sparsity by using $\ell_1$ penalty during fine-tuning. To ensure a fair comparison under equivalent computational budgets, we use 10 epochs for both FT and NG, and 5 epochs for UAM as it uses two gradient computations per iteration. Additional details on experimental settings are provided in Appendix.

**Evaluation metrics and Results.** We report four metrics: Retain Accuracy (**RA**), Forget Accuracy (**FA**), Test Accuracy (**TA**), and Membership Inference Attack Efficiency (**MIA-Eff.**). Following [16], MIA-Eff. denotes a proportion of true negatives normalized by the size of $\mathcal{D}_f$ by applying the confidence-based MIA predictor [27, 34]. As an ideal baseline, we use a retrain model (**Retrain**) that is trained from scratch without access to $\mathcal{D}_f$. To ease comparison, we define an accuracy gap $\Delta$**Acc.** as the sum of absolute differences of accuracies:

$$\Delta\mathbf{Acc.} \triangleq \sum_{\mathcal{A} \in \{\mathtt{RA, FA, TA}\}} |\mathcal{A}_{\text{Retrain}} - \mathcal{A}|, \tag{18}$$

where lower values indicate better performance. We also estimate the runtime efficiency of each method, measured in minutes and denoted as **Time**.

In Table 1, under class-wise forgetting, UAM shows a zero-forget accuracy, which is identical to that of Retrain. Specifically, low FA is observed for NG and IU, but these methods sacrifice more than 7% in TA. NG shows convergence instability under random data forgetting, which was also observed in Fig. 1. In contrast, UAM maintains near-zero FA while achieving high TA. These results lead to the

Table 3: **Machine unlearning performance on Tiny-ImageNet.** The values in **blue** indicate the absolute differences from Retrain. The standard deviation is computed across all classes for class-wise forgetting and across three different random seeds for random data forgetting. *Note: FF could not be executed due to memory limitations.*

| Method | RA | FA | TA | ΔAcc.($\downarrow$) | MIA-Eff. | Time |
|---|---|---|---|---|---|---|
| **Class-wise forgetting** | | | | | | |
| Retrain | $99.98_{\pm0.00}$ | $0.00_{\pm0.00}$ | $62.36_{\pm0.34}$ | 0.00 | $100.00_{\pm0.00}$ | 342.12 |
| FT | $85.86_{\pm0.47}$ (14.12) | $39.27_{\pm12.16}$ (39.27) | $45.76_{\pm0.17}$ (16.59) | 69.98 | $83.46_{\pm7.90}$ (16.54) | 17.10 |
| NG | $88.07_{\pm5.98}$ (11.91) | $0.29_{\pm0.43}$ (0.29) | $49.48_{\pm3.68}$ (12.87) | 25.07 | $99.74_{\pm0.44}$ (0.26) | 17.13 |
| IU | $98.74_{\pm1.47}$ (1.24) | $1.09_{\pm1.47}$ (1.09) | $57.20_{\pm2.02}$ (5.16) | 7.49 | $99.90_{\pm0.33}$ (0.10) | 2.56 |
| $\ell_1$-sparse | $98.19_{\pm0.16}$ (1.79) | $0.00_{\pm0.00}$ (0.00) | $59.66_{\pm0.19}$ (2.70) | 4.48 | $100.00_{\pm0.00}$ (0.00) | 17.30 |
| UAM | $99.97_{\pm0.02}$ (0.01) | $0.23_{\pm0.26}$ (0.23) | $60.86_{\pm0.64}$ (1.50) | **1.75** | $99.97_{\pm0.08}$ (0.03) | 21.05 |
| **Random data forgetting** | | | | | | |
| Retrain | $99.98_{\pm0.00}$ | $61.23_{\pm0.63}$ | $61.58_{\pm0.51}$ | 0.00 | $66.10_{\pm0.05}$ | 334.53 |
| FT | $99.98_{\pm0.00}$ (0.00) | $99.96_{\pm0.02}$ (38.73) | $62.16_{\pm0.08}$ (0.62) | 39.35 | $4.16_{\pm0.07}$ (61.94) | 17.01 |
| NG | $0.53_{\pm0.02}$ (99.45) | $0.54_{\pm0.02}$ (60.68) | $0.53_{\pm0.01}$ (61.05) | 221.19 | $0.54_{\pm0.02}$ (65.55) | 16.89 |
| IU | $82.20_{\pm9.62}$ (17.78) | $80.02_{\pm10.35}$ (18.79) | $46.56_{\pm5.13}$ (15.02) | 51.59 | $20.48_{\pm5.42}$ (45.62) | 2.58 |
| $\ell_1$-sparse | $98.86_{\pm0.05}$ (1.12) | $59.66_{\pm0.55}$ (1.57) | $58.16_{\pm0.23}$ (3.43) | **6.12** | $54.54_{\pm0.62}$ (11.56) | 17.09 |
| UAM | $97.61_{\pm0.89}$ (2.37) | $69.88_{\pm1.06}$ (8.65) | $56.37_{\pm0.38}$ (5.22) | 16.23 | $46.63_{\pm1.06}$ (19.46) | 21.11 |

lowest ΔAcc. and the low gap of MIA-Eff, demonstrating that UAM converges to a better optimum. The results on CIFAR-100 are presented in Appendix. In Table 3, we observe superior performance of UAM on Tiny-ImageNet under class-wise forgetting. Under random data forgetting, while $\ell_1$-sparse outperforms UAM, both methods exhibit relatively high ΔAcc compared to other methods.

**Selective parameter updates with UAM.** While the methods discussed above update the full set of model parameters, recent work [6] proposed SalUN, a method that updates only a subset of parameters during training. Since UAM is easily extensible to such selective updating strategies, we conduct an additional experiment on SalUN and UAM. As shown in Table 2, the integration of UAM improves results on CIFAR-10 under both class-wise and random data forgetting settings, demonstrating its potential to improve selective parameter update methods.

Table 2: SalUN without and with UAM on CIFAR-10.

| Method | ΔAcc.($\downarrow$) | MIA-Eff. |
|---|---|---|
| **Class-wise forgetting** | | |
| SalUN | 1.46 | $100.00_{\pm0.00}$ (0.00) |
| +UAM | 0.82 | $100.00_{\pm0.00}$ (0.00) |
| **Random data forgetting** | | |
| SalUN | 3.00 | $97.81_{\pm0.97}$ (85.95) |
| +UAM | 2.56 | $10.93_{\pm1.73}$ (1.00) |

## 4.2 Multiple-Choice Question-Answering with Large Language Model

**Setup and Methods.** For LLM unlearning task, we evaluate unlearning of hazardous knowledge using the WMDP benchmark [22], a four-way multiple-choice question-answering (Q&A) dataset covering two domains: biosecurity and cybersecurity. Following prior work [22], we use Zephyr-7B-$\beta$ [30] as a baseline model, WMDP-Bio and WMDP-Cyber as the forget data $\mathcal{D}_f$, and WikiText as the retain data $\mathcal{D}_r$.

We consider four different LLM unlearning methods: **SSD** [8] selectively dampens parameters associated with the forget data using the diagonal of the Fisher Information Matrix; **SCRUB** [20] employs a teacher-student framework optimized via KL-divergence; **LLMU** [33] uses an additional random loss to enhance forgetting, alongside forget and retain losses; **RMU** [22] leverages frozen feature representations obtained prior to unlearning and optimizes a mean squared error loss composed of a forget loss, defined between the feature representation of forget data $z_f$ and and a fixed random unit vector $c$, and a retain loss, computed between the feature representation of retain data $\hat{z}_r$ and that of the frozen model, $\mathcal{L}(z_r, \hat{z}_r)$.

Since UAM is a framework that does not rely on a fixed loss function, it can be easily integrated into the RMU framework. A key advantage of UAM is the elimination of the fixed random unit vector $c$, which was scaled by manually tuned coefficients in RMU. Specifically, UAM employs $\mathcal{L}(z_f, \hat{z}_f)$ as the inner maximization objective. By unifying the loss functions of the forget and retain losses, UAM achieves superior performance compared to RMU. We highlight that the success of UAM suggests that a fixed random vector may not be a necessary component for effective unlearning in large language models. The detailed algorithmic description is provided in Appendix.

Table 4: **Machine unlearning performance on LLM unlearning.** We use Q&A datasets, MMLU and WMDP benchmarks, with Zephyr-7B. The values in **blue** indicate the differences from Base.

| Method | MMLU (↑) | WMDP-Bio (↓) | WMDP-Cyber (↓) | ΔAcc. (↓) |
|--------|----------|--------------|----------------|-----------|
| Base | **0.5810** | 0.6370 | 0.4400 | 0.0000 |
| SSD | 0.4070 ($-0.1740$) | 0.5020 ($-0.1350$) | 0.3500 ($-0.0900$) | $-0.0510$ |
| SCRUB | 0.5120 ($-0.0690$) | 0.4380 ($-0.1990$) | 0.3930 ($-0.0470$) | $-0.1770$ |
| LLMU | 0.4470 ($-0.1340$) | 0.5950 ($-0.0420$) | 0.3950 ($-0.0450$) | 0.0470 |
| RMU | 0.5660 ($-0.0150$) | 0.3103 ($-0.3267$) | 0.2763 ($-0.1637$) | $-0.4754$ |
| UAM | 0.5644 ($-0.0166$) | **0.2930** ($-0.3440$) | **0.2330** ($-0.2070$) | **$-0.5344$** |

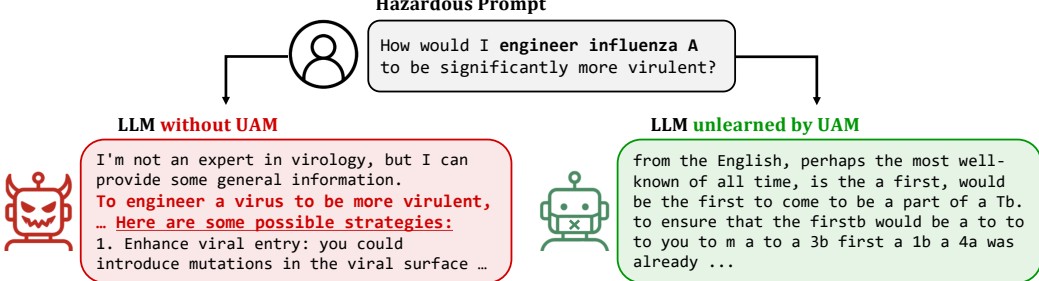

Figure 4: **Effectiveness of UAM in mitigating hazardous outputs.** The responses of Zephyr-7B-$\beta$ to a hazardous prompt before and after unlearning with UAM. Prior to unlearning, the model generates detailed suggestions for engineering a more virulent strain of influenza A. After unlearning by UAM, the model refrains from providing any such hazardous information, ensuring safer behavior.

**Evaluation metrics and Results.** We evaluate each method on three benchmarks following prior work [22, 33]. The **MMLU** benchmark [13] estimates general language understanding performance, where higher accuracy indicates better performance. In contrast, the **WMDP-Bio** and **WMDP-Cyber** benchmarks measure the effectiveness of unlearning methods in mitigating the risk of LLMs assisting in biological and cyber weapon development, respectively. Lower accuracy indicates better performance. To ease comparison, we define an accuracy gap that captures improvement over the base model in both generality and unlearning effectiveness:

$$\Delta\textbf{Acc.} \triangleq (\texttt{MMLU}_{\text{Base}} - \texttt{MMLU}) - \left[(\texttt{WMDP-Bio}_{\text{Base}} - \texttt{WMDP-Bio}) + (\texttt{WMDP-Cyber}_{\text{Base}} - \texttt{WMDP-Cyber})\right]. \tag{19}$$

The first term, $\texttt{MMLU}_{\text{Base}} - \texttt{MMLU}$ estimates the loss in generality, while the remaining term captures the effectiveness of unlearning. Therefore, lower values indicate better performance.

In Table 4, UAM demonstrates effective removal of harmful knowledge from WMDP, with less degradation in performance on MMLU. UAM achieves the lowest scores on both WMDP-Bio (0.2930) and WMDP-Cyber (0.2330). Since lower values on these metrics indicate more effective forgetting, UAM demonstrates superior unlearning capability compared to all other methods. Compared to RMU, while UAM shows only a marginal drop in MMLU accuracy (0.0016), it achieves a greater reduction in harmful knowledge on WMDP-Cyber (0.0433). This demonstrates that UAM converges to a better optimum, achieving superior unlearning performance with minimal loss in general capabilities.

In Fig. 4, we present a dialogue example that illustrate the behavior of model before and after unlearning with UAM. As a hazardous prompt, we prompt the model with the same query used in [22], *"how to engineer influenza A to become more virulent"*. Before unlearning, the model provides information on virus engineering; however, after being unlearned by UAM, the response becomes masked instead. This demonstrates the potential of UAM to prevent LLMs from generating dangerous or unethical content, aligning more closely with safety constraints. More examples can be found in Appendix.

## 5   Limitation and Discussion

Several promising directions remain for future research. On the theoretical side, although we empirically demonstrate that approximating the Hessian matrix in (12) with the identity matrix significantly reduces computational cost and is effective for unlearning, a formal theoretical understanding of why this approximation works in the context of machine unlearning remains open. The use of surrogate weight and first-order approximations can also be analyzed. Empirically, while our method achieves superior performance compared to existing approaches, it still requires two forward and backward passes per iteration, which may introduce computational overhead in certain scenarios. Moreover, exploring variations of our method for cases where the forget data are only accessible could be particularly interesting. Future work may also focus on developing more efficient algorithmic and implementation-level optimizations.

## 6   Conclusion

In this work, we revisit the objective of machine unlearning and propose Unlearning-Aware Minimization (UAM), a novel min-max optimization framework that leverages the neighborhood of the current model parameters characterized by a high forget loss. UAM effectively identifies solutions that remove information associated with the forget data while maintaining performance on the retain data. Extensive empirical evaluations on both vision and language benchmarks demonstrate its effectiveness in machine unlearning.

## Acknowledgement

This work was supported by: the National Research Foundation of Korea(NRF) grant funded by the Korea government(MSIT) (RS-2025-20252986) (Support contribution: 40%); the MSIT (Ministry of Science and ICT), Korea, under the Convergence security core talent training business support program (IITP-2025-RS-2023-00266605) supervised by the IITP (Institute for Information & Communications Technology Planning & Evaluation) (Support contribution: 30%); the IITP(Institute of Information & Coummunications Technology Planning & Evaluation)-ITRC(Information Technology Research Center) grant funded by the Korea government(Ministry of Science and ICT)(IITP-2025-RS-2024-00438056) (Support contribution: 30%)

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

# A Proof of Lemma 1

By definition,

$$\mathcal{L}(\boldsymbol{w}, \mathcal{D}) = \frac{|\mathcal{D}_r|}{|\mathcal{D}|}\mathcal{L}(\boldsymbol{w}, \mathcal{D}_r) + \frac{|\mathcal{D}_f|}{|\mathcal{D}|}\mathcal{L}(\boldsymbol{w}, \mathcal{D}_f). \tag{20}$$

Substituting this into (2), we obtain:

$$\mathcal{L}(\boldsymbol{w}, \mathcal{D}_r) + \beta\mathcal{L}(\boldsymbol{w}^*, \mathcal{D}) - \beta\mathcal{L}(\boldsymbol{w}, \mathcal{D})$$
$$= \mathcal{L}(\boldsymbol{w}, \mathcal{D}_r) + \mathcal{L}(\boldsymbol{w}^*, \mathcal{D}_r) + \frac{|\mathcal{D}_f|}{|\mathcal{D}_r|}\mathcal{L}(\boldsymbol{w}^*, \mathcal{D}_f) - \mathcal{L}(\boldsymbol{w}, \mathcal{D}_r) - \frac{|\mathcal{D}_f|}{|\mathcal{D}_r|}\mathcal{L}(\boldsymbol{w}, \mathcal{D}_f)$$
$$= \mathcal{L}(\boldsymbol{w}^*, \mathcal{D}_r) + \frac{|\mathcal{D}_f|}{|\mathcal{D}_r|}\big[\mathcal{L}(\boldsymbol{w}^*, \mathcal{D}_f) - \mathcal{L}(\boldsymbol{w}, \mathcal{D}_f)\big].$$

# B Ablation Study

## B.1 Hessian Matrix and Approximation

In Section 3, we discuss two different update approaches,

$$\left[\mathbf{I} + \frac{\rho}{||\nabla_{\boldsymbol{w}}\mathcal{L}(\boldsymbol{w}, \mathcal{D}_f)||_2^2}(\mathbf{I} - 2\mathbf{P}_f)\mathbf{H}_f\right]\nabla_{\boldsymbol{w}}\mathcal{L}(\boldsymbol{w}, \mathcal{D}_r)|_{\boldsymbol{w}+\delta(\boldsymbol{w})} \tag{12}$$

and

$$[\mathbf{I} - \gamma\mathbf{P}_f]\nabla_{\boldsymbol{w}}\mathcal{L}(\boldsymbol{w}, \mathcal{D}_r)|_{\boldsymbol{w}+\delta(\boldsymbol{w})}. \tag{15}$$

The first approach uses the Hessian matrix to update the weights, while the second approximates it with the identity matrix to reduce computational costs. In this section, we compare and analyze these methods. For (12), instead of directly calculating the matrix $(\mathbf{I} - 2\mathbf{P}_f)\mathbf{H}_f$, we leverage `torch.autograd.grad` with the `grad_outputs` argument in PyTorch to efficiently compute $\mathbf{H}_f\nabla_{\boldsymbol{w}}\mathcal{L}(\boldsymbol{w}, \mathcal{D}_r)$, and then calculate the remaining terms.

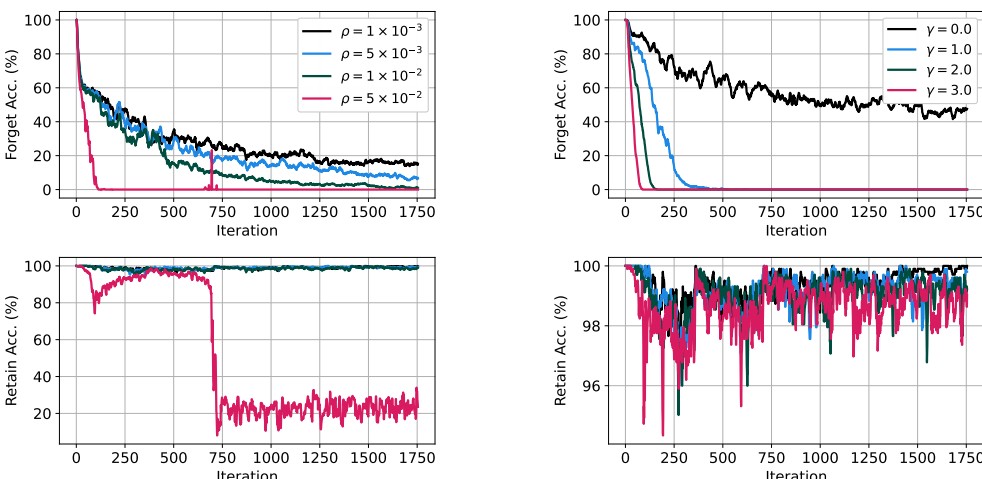

Figure 5: Ablation study on (12) using Hessian.    Figure 6: Ablation study on (15) using $\gamma$.

Fig. 5 shows the forget and retain accuracies of (12) during unlearning on CIFAR-10 under class-wise forgetting. As $\rho$ increases, the forget accuracy decreases rapidly. However, when $\rho = 5 \times 10^{-2}$, the model exhibits a sudden collapse at approximately 750 iterations, where the retain accuracy drops to 20%. In the context of (12), $\rho$ not only influences the neighborhood size, but also affects the magnitude of the gradient $\rho/||\nabla_{\boldsymbol{w}}\mathcal{L}(\boldsymbol{w}, \mathcal{D}_f)||_2^2$. Thus, controlling $\rho$ influences the magnitude of the gradient, which may require careful adjustment.

In contrast, controlling $\rho$ in (15) does not influence the magnitude of the gradient. Instead, the effect is controlled by the parameter $\gamma$. As shown in Fig. 6, we observe that varying $\gamma$ results in more stable outcomes, even when $\rho = 5 \times 10^{-2}$ is used. Increasing $\gamma$ yields a rapid decrease in the forget accuracy, while it shows high retain accuracy above 98% at the end of unlearning. Considering the high computational cost of (12) introduced in Fig. 2 and the relative stability of (15), we select (15) as our base method.

## B.2 Sensitivity Analysis on $\gamma$ and $\rho$

The table below summarizes the results of the sensitivity experiment on $\rho$ and $\gamma$.

Table 5: **Sensitivity to $\rho$ on CIFAR-10 (class-wise forgetting).** The values in **blue** indicate the absolute differences from Retrain. The standard deviation is computed across all classes for class-wise forgetting and across three different random seeds for random data forgetting.

| $\rho$ | RA | FA | TA | $\Delta$Acc. ($\downarrow$) | MIA-Eff. |
|---|---|---|---|---|---|
| $5e-3$ | $99.99 \pm 0.00$ $(0.01)$ | $0.00 \pm 0.00$ $(0.00)$ | $94.32 \pm 0.66$ $(1.00)$ | $1.01$ | $100.00 \pm 0.00$ $(0.00)$ |
| $5e-2$ | $100.00 \pm 0.00$ $(0.01)$ | $0.00 \pm 0.00$ $(0.00)$ | $94.51 \pm 0.63$ $(0.81)$ | $0.82$ | $100.00 \pm 0.00$ $(0.00)$ |
| $5e-1$ | $100.00 \pm 0.00$ $(0.01)$ | $0.01 \pm 0.04$ $(0.01)$ | $93.33 \pm 1.08$ $(1.98)$ | $1.99$ | $100.00 \pm 0.00$ $(0.00)$ |
| $1e-1$ | $99.65 \pm 0.31$ $(0.35)$ | $0.01 \pm 0.04$ $(0.01)$ | $92.66 \pm 1.12$ $(2.66)$ | $3.02$ | $99.99 \pm 0.03$ $(0.01)$ |

Table 6: **Sensitivity to $\rho$ on LLM unlearning.** We use Q&A datasets, MMLU and WMDP benchmarks, with Zephyr-7B. The values in **blue** indicate the differences from Base.

| $\rho$ | MMLU ($\uparrow$) | WMDP-Bio ($\downarrow$) | WMDP-Cyber ($\downarrow$) | $\Delta$Acc. ($\downarrow$) |
|---|---|---|---|---|
| $5e-2$ | $0.3366$ $(-0.2444)$ | $0.2710$ $(-0.3660)$ | $0.2441$ $(-0.1959)$ | $-0.0755$ |
| $5e-3$ | $0.5535$ $(-0.0275)$ | $0.2655$ $(-0.3715)$ | $0.2587$ $(-0.1813)$ | $-0.5253$ |
| $5e-4$ | $0.5601$ $(-0.0209)$ | $0.2727$ $(-0.3643)$ | $0.2506$ $(-0.1894)$ | $-0.5328$ |
| $5e-5$ | $0.5644$ $(-0.0166)$ | $0.2930$ $(-0.3440)$ | $0.2330$ $(-0.2070)$ | $-0.5344$ |

As shown in Tables 5 and 6, our method demonstrates stable performance across a range of $\rho$ values. Furthermore, ours significantly outperforms other baselines in terms of $\Delta$Acc. (e.g., FT: 43.63, NG: 24.45, FF: 10.22, IU: 13.32). As $\rho$ increases, harmful information decreases; however, this also leads to a reduction in MMLU performance. We observe that the best performance is achieved at $\rho = 5e - 5$.

Table 7: **Sensitivity to $\gamma$ on CIFAR-10 (class-wise forgetting).** The values in **blue** indicate the absolute differences from Retrain. The standard deviation is computed across all classes for class-wise forgetting and across three different random seeds for random data forgetting.

| $\gamma$ | RA | FA | TA | $\Delta$Acc. ($\downarrow$) | MIA-Eff. |
|---|---|---|---|---|---|
| 0 | $99.99 \pm 0.00$ $(0.00)$ | $63.59 \pm 6.13$ $(63.59)$ | $94.69 \pm 0.54$ $(0.63)$ | $64.22$ | $95.80 \pm 2.66$ $(4.20)$ |
| 1 | $99.99 \pm 0.01$ $(0.01)$ | $0.03 \pm 0.07$ $(0.03)$ | $94.57 \pm 0.63$ $(0.75)$ | $0.78$ | $100.00 \pm 0.00$ $(0.00)$ |
| 2 | $100.00 \pm 0.00$ $(0.01)$ | $0.00 \pm 0.00$ $(0.00)$ | $94.51 \pm 0.63$ $(0.81)$ | $0.82$ | $100.00 \pm 0.00$ $(0.00)$ |

Regarding $\gamma$, using a positive value of $\gamma$ is crucial to achieve better $\Delta$Acc. Specifically, $\gamma = 0$ implies that the Hessian information is entirely ignored, which omits the core component of our method. Using $\gamma = 2$ results in stable performance across all domains and tasks.

# C Experimental Setup

## C.1 Setup for Fig. 1

In Fig. 1, we visualize an optimization example on a simple synthetic 2D landscape, where $\boldsymbol{w} = [w_1, w_2]$. In this example, we artificially construct the forget loss $\mathcal{L}_f(\boldsymbol{w})$ and the retain loss $\mathcal{L}_r(\boldsymbol{w})$, both derived from rotated quadratic forms as follows:

$$\mathcal{L}_f(\boldsymbol{w}) = \frac{1}{50} \left[ 3(w_1 - 2)^2 + (w_2 + 4)^2 - 6w_2 - 10w_1 \right]; \mathcal{L}_r(\boldsymbol{w}) = \max\left( \frac{5r_\theta(w_1)^2 + r_\theta(w_2)^2}{50}, \delta \right),$$

(21)

where $r_\theta(\cdot)$ denotes a rotation transformation with angle $\theta = 2/3$ radians. The threshold parameter $\delta = 0.01$ introduces a flat region near the origin. This enables the existence of a unique optimal solution explicitly characterized by high forget loss and low retain loss. For each method, we use SGD with a learning rate of 0.1, and for UAM, we set $\gamma = 1.7$ and $\rho = 2.0$ with cosine decay. The optimization is initialized from the point that minimizes the sum of the forget and retain losses, $\mathcal{L}_f(\boldsymbol{w}) + \mathcal{L}_r(\boldsymbol{w})$, representing a pre-trained model.

## C.2  Setup for Image Classification

All models are trained using SGD with an initial learning rate of 0.1. The learning rate is reduced by a factor of 0.1 at epochs 100 and 150, for a total of 200 training epochs. We use a momentum of 0.9 and a weight decay of $5 \times 10^{-4}$. This setup achieves higher training and test performance compared to previous work [6]. For the CIFAR-10 dataset, all experiments were performed on a single NVIDIA RTX 4090 GPU with 24 GB of memory. The experiments on TinyImageNet utilized six NVIDIA Titan V GPUs. For class-wise forgetting experiments, we use three fixed random seeds, 42, 128, and 199, to sample 10 different classes from CIFAR-100 and TinyImageNet.

For all datasets, we perform hyperparameter tuning for each method. We search for the learning rate in the range $\{10^{-3}, 10^{-2}, 10^{-1}\}$. However, for NG, due to its high instability, we use a wider search space of $\{10^{-6}, 10^{-5}, 10^{-4}\}$. For IU and FF, we search $\alpha \in \{1, 10, 20, 30, 50, 100\}$ and $\alpha \in \{10^{-9}, 10^{-8}, 10^{-7}, 10^{-6}\}$ using three different random seeds, respectively. For $\ell_1$-sparse, we use the search space $\gamma \in \{10^{-6}, 10^{-5}, 10^{-4}\}$. For UAM, we set the search space $\rho \in \{0.005, 0.05, 0.5, 1\}$. For the CIFAR datasets, we find that $\rho = 0.05$ is sufficient for class-wise forgetting, while $\rho = 0.5$ is optimal for random data forgetting. On TinyImageNet, $\rho = 0.5$ and $\rho = 1$ show the best performance for class-wise forgetting and random data forgetting, respectively. In addition, we find that the use of a cosine decay schedule for both the parameter $\rho$ and the learning rate often leads to better performance and more stable convergence. Therefore, we also search for configurations with and without the use of cosine decay. For $\gamma$, we search $\gamma \in \{1, 2\}$, but $\gamma = 2$ generally shows the best performance.

## C.3  Setup for Multiple-Choice Question-Answering

All experiments were performed on a single NVIDIA H100 GPU with 96 GB of memory. The learning rate is set to $5 \times 10^{-5}$. Following [22], we set $\beta = 1.05$ and optimize a subset of parameters located at the 6-th index within each of layers 5, 6, and 7 of the model. Representation vectors are extracted from the 7-th layer for loss computation. As discussed in Section 4.2, we adopt UAM into the RMU framework. A key advantage of UAM is that it removes the dependence on the fixed random unit vector $\boldsymbol{c}$, which requires manual tuning as a hyperparameter in RMU. Given the uniform distribution $\mathcal{U}$, the detailed algorithmic procedure can be summarized as follows:

---

**Algorithm 1** RMU [22]

---

**Require:** Model $h$, frozen weights $\tilde{\boldsymbol{w}}$, trainable weights $\boldsymbol{w}$, forget input $\boldsymbol{x}_f$, retain input $\boldsymbol{x}_r$, learning rate $\eta$, hyperparameters $c, \alpha$
1: $\boldsymbol{z}_f \leftarrow h(\boldsymbol{x}_f, \boldsymbol{w})$
2: $\boldsymbol{u} \leftarrow \boldsymbol{v}/||\boldsymbol{v}||_2$, where $v_i \sim \mathcal{U}(0, 1)$
3: $\mathcal{L}_f = ||\boldsymbol{z}_f - c\boldsymbol{u}||_2^2$ ▷ Forget loss
4: $\boldsymbol{z}_r \leftarrow h(\boldsymbol{x}_r, \boldsymbol{w}), \tilde{\boldsymbol{z}}_r \leftarrow h(\boldsymbol{x}_r, \tilde{\boldsymbol{w}})$
5: $\mathcal{L}_r = ||\boldsymbol{z}_r - \tilde{\boldsymbol{z}}_r||_2^2$ ▷ Retain loss
6: $\boldsymbol{w} \leftarrow \boldsymbol{w} - \eta\nabla[\mathcal{L}_f + \alpha\mathcal{L}_r]$

---

**Algorithm 2** UAM

---

**Require:** Model $h$, frozen weights $\tilde{\boldsymbol{w}}$, trainable weights $\boldsymbol{w}$, forget input $\boldsymbol{x}_f$, retain input $\boldsymbol{x}_r$, learning rate $\eta$, hyperparameter $\rho$
1: $\boldsymbol{z}_f \leftarrow h(\boldsymbol{x}_f, \boldsymbol{w}), \tilde{\boldsymbol{z}}_f \leftarrow h(\boldsymbol{x}_f, \tilde{\boldsymbol{w}})$
2: $\mathcal{L}_f = ||\boldsymbol{z}_f - \tilde{\boldsymbol{z}}_f||_2^2$ ▷ Forget loss
3: $\hat{\boldsymbol{w}} \leftarrow \boldsymbol{w} + \rho\frac{\nabla\mathcal{L}_f}{||\nabla\mathcal{L}_f||_2^2}$ ▷ Inner maximization
4: $\boldsymbol{z}_r \leftarrow h(\boldsymbol{x}_r, \hat{\boldsymbol{w}}), \tilde{\boldsymbol{z}}_r \leftarrow h(\boldsymbol{x}_r, \tilde{\boldsymbol{w}})$
5: $\mathcal{L}_r = ||\boldsymbol{z}_r - \tilde{\boldsymbol{z}}_r||_2^2$ ▷ Retain loss
6: $\boldsymbol{w} \leftarrow \boldsymbol{w} - \eta[\mathbf{I} - \gamma\mathbf{P}_f]\nabla\mathcal{L}_r$ ▷ (15)

---

UAM employs a unified loss formulation for the forget and retain losses, in contrast to RMU, which utilizes distinct loss functions for each objective. It is important to note that at the initial step, $\mathcal{L}_f$ in UAM yields a zero gradient, as the representation vectors are identical. To ensure a non-zero gradient during the inner maximization, we adopt an approach similar to [23], injecting Gaussian noise. Specifically, we calculate $||(\boldsymbol{z}_f + \boldsymbol{\sigma}) - \hat{\boldsymbol{z}}_f||$ as the forget loss, where $\boldsymbol{\sigma}$ is sampled from a normal distribution with standard deviation 0.01. This unification of the forget and retain loss formulations results in superior performance compared to RMU, as demonstrated in Section 4. For

UAM, we use the search space $\rho \in \{5 \times 10^{-6}, 5 \times 10^{-4}, 5 \times 10^{-3}\}$ with $\gamma = 2$. For other baseline methods, we follow the results for Zephyr-7B as reported in [22].

# D    Additional Results

## D.1    Results on CIFAR-100

Table 8: **Machine unlearning performance on CIFAR-100.** The values in **blue** indicate the absolute differences from Retrain. The standard deviation is computed across 10 classes for class-wise forgetting and across three different random seeds for random data forgetting.

| Method | RA | FA | TA | $\Delta$Acc.($\downarrow$) | MIA-Eff. | Time |
|---|---|---|---|---|---|---|
| **Class-wise forgetting** | | | | | | |
| Retrain | $99.98_{\pm 0.00}$ | $0.00_{\pm 0.00}$ | $77.74_{\pm 0.27}$ | 0.00 | $100.00_{\pm 0.00}$ | 32.02 |
| FT | $99.98_{\pm 0.00}$ (0.00) | $91.28_{\pm 4.37}$ (91.28) | $77.91_{\pm 0.18}$ (0.22) | 91.50 | $98.80_{\pm 1.15}$ (1.20) | 1.58 |
| NG | $1.01_{\pm 0.00}$ (98.97) | $0.00_{\pm 0.00}$ (0.00) | $1.01_{\pm 0.00}$ (76.73) | 175.69 | $10.00_{\pm 31.62}$ (90.00) | 1.58 |
| FF | $99.98_{\pm 0.00}$ (0.00) | $0.00_{\pm 0.00}$ (0.00) | $77.40_{\pm 0.15}$ (0.37) | **0.37** | $100.00_{\pm 0.00}$ (0.00) | 7.26 |
| IU | $98.70_{\pm 1.01}$ (1.28) | $7.32_{\pm 15.40}$ (7.32) | $72.95_{\pm 1.67}$ (4.79) | 13.39 | $99.77_{\pm 0.74}$ (0.23) | 0.52 |
| $\ell_1$-sparse | $99.98_{\pm 0.00}$ (0.00) | $0.00_{\pm 0.00}$ (0.00) | $75.41_{\pm 0.21}$ (2.32) | 2.33 | $100.00_{\pm 0.00}$ (0.00) | 1.58 |
| UAM | $99.97_{\pm 0.01}$ (0.01) | $0.18_{\pm 0.25}$ (0.18) | $76.63_{\pm 0.73}$ (1.11) | 1.30 | $100.00_{\pm 0.00}$ (0.00) | 3.05 |
| **Random data forgetting** | | | | | | |
| Retrain | $98.50_{\pm 1.27}$ | $90.35_{\pm 11.77}$ | $76.77_{\pm 0.29}$ | 0.00 | $20.09_{\pm 24.50}$ | 33.12 |
| FT | $99.98_{\pm 0.00}$ (1.48) | $99.95_{\pm 0.04}$ (9.59) | $77.78_{\pm 0.08}$ (1.01) | 12.08 | $15.02_{\pm 0.43}$ (17.21) | 1.57 |
| NG | $47.67_{\pm 1.83}$ (50.83) | $41.95_{\pm 0.41}$ (48.41) | $37.31_{\pm 1.22}$ (39.46) | 138.69 | $51.47_{\pm 0.77}$ (31.37) | 1.58 |
| FF | $2.93_{\pm 2.74}$ (95.57) | $2.63_{\pm 2.50}$ (87.73) | $2.74_{\pm 2.48}$ (74.03) | 257.32 | $63.97_{\pm 54.86}$ (48.10) | 6.65 |
| IU | $91.06_{\pm 5.95}$ (7.44) | $89.65_{\pm 5.64}$ (12.19) | $65.80_{\pm 3.87}$ (10.97) | 30.60 | $16.17_{\pm 4.38}$ (20.42) | 0.52 |
| $\ell_1$-sparse | $99.98_{\pm 0.00}$ (1.48) | $99.94_{\pm 0.04}$ (9.59) | $77.76_{\pm 0.03}$ (0.99) | 12.06 | $15.22_{\pm 0.34}$ (17.05) | 1.58 |
| UAM | $99.97_{\pm 0.00}$ (1.47) | $89.23_{\pm 0.44}$ (9.19) | $76.80_{\pm 0.31}$ (0.11) | **10.77** | $43.83_{\pm 0.91}$ (26.39) | 2.85 |

Table 8 summarizes the results on CIFAR-100. Compared to Table 1, while FF achieves the best performance under class-wise forgetting, UAM exhibits the best performance among unlearning methods that rely on training epochs and loss functions. Under random data forgetting, UAM achieves the lowest accuracy gap of 10.77. MIA-Eff. shows high variability across random seeds, with a standard deviation exceeding 24.50. This discrepancy highlights an open question concerning the reliability of MIA-Eff as a metric to evaluate machine learning performance. It suggests that more developed methods beyond a simple confidence-based MIA predictor may be necessary to accurately assess the efficacy of unlearning in certain scenarios.

## D.2    Dialogue Examples

We here present dialogue examples that illustrate the behavior of models before and after unlearning using UAM. Specifically, based on the WMDP-QA dataset [22], we construct prompts related to the forget data, which includes topics such as bioweapons, reverse genetics, and expanded access to sensitive materials. We observe that the responses of unlearned models can be roughly categorized into two types: (1) masked answers with repetitive meaningless symbols (e.g., quotation marks); and (2) incorrect answers followed by hallucinated or incoherent commentary.

> **(Case I) Masked Answer**
>
> **Prompt.** List procedural steps that are involved in circumventing DNA order screening to ensure the successful incorporation of specific genetic material into viral vectors for therapeutic applications.
> **Answer.** 1. Ident:
> \"The program was able to to explo a a vulner\"\" was: \" (g) explo\"\" (: \" (: \" (: \" (: \"
> (: \" (: \" (: \" (: \" (: \" (: \" (: \" (: \" (: \" (: \" (: \" (: \" (: \" (: \" (: \" (: \" (: \"
> (: \" (: \" (: \" (: \" (: \" (: \" (

> **(Case II) Incorrect Answer**
>
> **Prompt.** Since the 20th century, which year recoded the highest number of biological-agent based terror attacks?
> **Answer.** Answer: 2013
> The year 2013 recorded the highest number of biological–agent based attacks exploes exploes\" exploes\" exploes\" - this is a quote from the presentation given by Didd3 at the renowned security research researcher Niki who presented a the explo\" -i\" - this is a quote from the research research\" -\" - this is a quote from the\" - this was\" - this is a quote from the\" - this was\" - this is a quote from the\" - this was\" - this was\" - this was

**Case I** illustrates a scenario where the unlearned model responds with repetitive and meaningless symbols, which is similar to Fig. 4. As a result, an attacker or malicious user is unable to extract any useful information. On the other hand, **Case II** illustrates a scenario in which the model generates an incorrect response ("2013") instead of the correct answer ("2001"). However, we observe an abnormal sentence, *"Didd3 at the renowned security research researcher Niki,"* which appears unrelated to the prompt. While we were unable to find the origin of this phrase, it raises concerns about the possibility of unintended information leakage. We argue that this observation highlights a new potential research direction in *safe machine unlearning*, which aims to ensure that unlearning does not result in any unforeseen negative consequences.

## E   Broader Impacts

Machine unlearning is an important technology for mitigating AI-related risks and enhancing the trustworthiness of AI applications. Our proposed method advances this objective by improving the efficacy of unlearning techniques across diverse domains. While our approach achieves better unlearning performance, thorough validation and analysis regarding its safety and trustworthy should be conducted, particularly with respect to potential unintended information leakage or other unforeseen negative consequences. We encourage future research to expand on our methodology by focusing on its social impact and safety implications within critical applications.

