# OpenReview forum: "Unlearning-Aware Minimization"
_NeurIPS.cc/2025/Conference — NeurIPS 2025 poster_

### Official Review · Reviewer_QDnv · 2025-07-02

**Clarity:** 3
**Significance:** 3
**Originality:** 2
**Rating:** 4
**Confidence:** 5

**Summary:**

In this paper, authors focus on the machine unlearning problem, which aims to eliminate the influence of forget data while preserving the performance on the retain data. The article points out that solely using negative gradient or fine-tuning methods fail to effectively balance the objectives of maximizing forget loss and minimizing retain loss. And authors propose a novel min-max optimization framework to consistently achieve higher forget loss and lower retain loss. Based on first-order Taylor expansion, the paper proposes an efficient solution to the above optimization and theoretically analyzes the validity of the update strategy.

**Questions:**

- In the related work 2.1, the authors mention that “our method is a min-max optimization framework rather than relying on dual-objective formulations.” What are the advantages of min-max optimization framework compared to dual-objective formulations?
- From eq.3, it can be found that the variable in the minimization problem is w, which means the w* is fixed. And Eq.5 proposes a maximiaztion problem to estimate w*. Therefore, when we solve the maximization problem, the w* should be fix to w0+\rho (\nabla \nabla L_f(w))/(||\nabla \nabla L_f(w)||_2^2). However, in the update process of eq.7, the w* is still treated as a variable. It seems the problem in eq.3 is not equal to that in eq.7.
- Under random data setting, since the gradients calculated on retain data are similar to those calculated on forget data, it can be found that the results of UAM are relatively weaker, is there any solution to mitigate this problem?
- In practical large-scale settings (e.g., LLMs), users may only provide forget sets. Is it realistic or necessary to assume full access to both retain and forget data partitions?
- When using a first-order Taylor approximation, please specify the smoothness requirements for the function.
- The graphical explanation in Figure 1 contains a typo: “GA”.
- There is a grammatical error in line 113 on page 3: “where and Ω is a finite upper bound on the distance.”
- The following sentence in the abstract may be ambiguous and could benefit from further clarification:“In this paper, we propose Unlearning-Aware Minimization (UAM), a novel min-max optimization framework that identifies weights with high loss on the forget data and uses their gradients to minimize the loss on the retain data.”

**Ethical Concerns:**

["NO or VERY MINOR ethics concerns only"]

**Final Justification:**

The response has addressed my concerns. I tend to increase my score to 4.

**Limitations:**

yes

**Paper Formatting Concerns:**

NAN

**Quality:**

2

**Strengths And Weaknesses:**

Strength
- The key innovation of this paper lies in formulating the machine unlearning problem within a min-max optimization framework, which leads to promising empirical performance.
- By leveraging a first-order Taylor approximation, the authors transform a challenging bi-level optimization into a tractable update rule, enabling efficient gradient-based training.
- The paper provides an intuitive geometric interpretation, illustrating how the alignment between the gradients of forget and retain data affects the optimization process, thereby enhancing the clarity and interpretability of the proposed method.

Weakness:

While the paper presents a novel and empirically effective framework for machine unlearning, several important aspects deserve further clarification and theoretical grounding:
- About assumption and setting: The assumption that both the forget and retain data are known and accessible may not hold in realistic large-scale settings such as LLMs, where only the forget set is identifiable. This limits the method’s applicability in more practical scenarios.

- About the technique: The method relies on an approximate optimization using a surrogate objective. However, it remains unclear whether the solution trajectory truly converges to the intended optimal parameter w^*,and how large the approximation gap is between the proxy and the true formulation. The clarification of the real gap is necessary for readers to understand the method.

- About the theory for the effectiveness of model unlearning.
The effectiveness of forgetting (i.e., removing the influence of the forget set) and maintaining performance on the retain set is empirically verified, but lacks formal guarantees. Does the proposed objective ensure meaningful unlearning under general conditions? Is there a generalization bound or any guarantee of worst-case behavior?

---

> ### Author Rebuttal · Authors · 2025-07-31
>
> We appreciate the reviewer’s insightful evaluation of the strengths and weaknesses of our paper. We believe that your feedback provides valuable perspectives that will help enhance the quality and clarity of our work.
>
> Below, we provide a detailed response addressing each of the weaknesses [W#] and questions [Q#], with references to the main paper [#] as well as new supplementary references [S#].
>
> ---
>
> **[W1, Q4] Regarding the setting using both forget and retain data**
>
> We investigated all recent papers on LLM and Vision unlearning over the past two years from top-tier conferences (NeurIPS, ICML, ACL, EMNLP, NAACL, CVPR, ICCV). **In total, we reviewed 71 papers, a detailed list of which will be provided as a comment.** Among these, 39 papers focus primarily on approximate unlearning (excluding certified unlearning), which is also the main theme of our work.
>
> | **Approximate unlearning** | **Use Retain & Forget** | **Forget Only** | **Retain Only** | **Others** |
> | --- | --- | --- | --- | --- |
> | **#Papers** | 27 | 3 | 5 | 4 |
>
> Among these, 29 papers utilize both the retain and forget datasets. While access to both datasets may be limited in practical scenarios, we argue that it is a common assumption in the current research field to use both.
>
> Nonetheless, we agree that evaluating our method in a more restricted setting is valuable for broader applicability. Below, we conducted experiments with a reduced number of forget data for class-wise forgetting on CIFAR-10. Random data forgetting is more similar to instance-wise forgetting, which does not align with the limited forget data. We used only 10% of each removed class and compared methods that are influenced by adjustments to the forget dataset, namely NG and IU.
>
> | 10% | **RA** | **FA** | **TA** | **MIA-Eff.** |
> | --- | --- | --- | --- | --- |
> | NG | 94.05±3.28 | 21.09±1.41 | 87.96±3.31 | 1.10±20.70 |
> | IU | 99.93±0.15 | 81.20±14.54 | 95.01±0.83 | 19.55±55.21 |
> | UAM | 99.99±0.01 | 0.00±0.00 | 94.75±0.66 | 100.00±0.00 |
>
> Interestingly, despite reducing the size of the forget dataset, **our method still demonstrates high performance, whereas the performance of NG and IU degraded significantly.** In particular, IU shows high forget accuracy due to the smaller forget dataset, which leads to an inaccurate approximation of the weight adjustment direction. **We believe that this experiment effectively highlights the robustness of our method.** We sincerely appreciate your insightful comments and will ensure that the revised version addresses this phenomenon.
>
> ---
>
> **[W2, Q2] Regarding the optimization formula**
>
> According to [S7], approximate unlearning can be classified into two categories: strong unlearning and weak unlearning. Strong unlearning seeks to match the model parameters or their distribution exactly, while weak unlearning aims to produce outputs that are similar to those of a retrained model. Our work aligns with weak unlearning, rather than strong unlearning. As noted in other weak unlearning methods [6, 16, 20], accurately quantifying the gap between the original and retrained models is inherently difficult within this framework. For a more detailed discussion of the theoretical guarantees, we refer the reviewer to [W3]. In this context, the optimal parameter $w^*$ cannot be directly inferred under the weak unlearning framework. As a result, **we utilize a surrogate parameter, $\hat{w}$.** It is important to clarify that $\hat{w}$ is not a fixed parameter; rather, **it can be any point within the set $ \hat{w} \in B^{\epsilon}_{\Omega} (w_0; D_f) $**. **We argue that this surrogate parameter provides valuable information that can guide the model towards the optimal point $w^*$.** This continuous optimization process is closely related to concepts such as generalization [7, 22] and adversarial robustness [10, 29, 37]. We will revise this section to make these points clearer. Once again, thank you for your comment.
>
> ---
>
> **[W3] Lacks formal guarantees on generalization bound or any guarantee of worst-case behavior.**
>
> Our work focuses on approximate unlearning with weak unlearning [6, 16, 23], which addresses unlearning by empirically retraining models. **Theoretical guarantees, such as generalization bounds, are more commonly associated with certified unlearning [S1, S5, S6],** a distinct area that combines approximate unlearning with differential privacy and remains less explored in machine learning [S1].
>
> While approximate unlearning methods lack formal guarantees, they have gained significant attention (e.g., L1-sparse [16] with 156 citations, SalUN [6] with 189 citations, and RMU [23] with 219 citations). Some works have attempted theoretical guarantees for approximate unlearning algorithms ([S3], [S4]), but these usually offer only upper bounds on information retained in model weights, rather than strict guarantees. **This highlights the challenge in providing formal guarantees for non-certified unlearning.** As noted by [20], unlearning lacks a well-defined formal definition, complicating the development of standardized evaluation metrics. Furthermore, most theoretical work has focused on certified unlearning, emphasizing that formal guarantees for approximate unlearning remain an open problem.
>
> However, [S2] suggests that certified unlearning methods can be adapted to approximate unlearning, particularly for convex models, which could improve their applicability to deep neural networks. This points to a potential integration of both approaches. **We believe that our investigation into the Hessian contributes to advancing certified unlearning**. Thank you for your feedback.
>
> ---
>
> **[Q1] Regarding the statement in Section 2.1:**
>
> In the sentence, we would like to clarify that we have developed a novel unlearning algorithm based on min-max optimization. The statement was not meant to argue that the min-max optimization approach is superior to dual-objective formulations. We will revise this part of the manuscript to make this distinction clearer. Thank you for your detailed comment.
>
> ---
>
> **[Q3] Regarding similarity between forget and retain gradients**
>
> Thank you for your insightful suggestions. Inspired by your idea, we first investigate the cosine similarity between the forget gradient, $\nabla L(w, D_f)$, and the retain gradient, $\nabla L(w, D_r)$, under random-data forgetting. To do this, with a fixed random seed (seed=42), we observe that the average cosine similarity is higher under random-data forgetting than under class-wise forgetting. Specifically, the cosine similarity in random-data forgetting is greater than that in class-wise forgetting. **However, it is noteworthy that the cosine similarity does not approach 1.** Instead, it fluctuates between -0.25 and 0.50.
>
> Under the random-data forgetting setting, **we also observe that our method successfully disentangles the gradients.** The table below illustrates the range of cosine similarity across iterations during our unlearning:
>
> | Iteration | 0-500 | 500-1000 | 1000-1500 |
> | --- | --- | --- | --- |
> | Cosine value range | [-0.30,0.50] | [-0.25,0.25] | [0.00, 0.15] |
>
> Therefore, our method shows its effectiveness in identifying implicit features where the forget and retain gradients can be separated. This suggests that our method is particularly useful when it is possible to distinguish between features to be forgotten and those to be retained.
>
> Beyond this result, the poor performance on random-data forgetting is a common phenomenon in machine unlearning [6, 16], as it is more challenging due to the high similarity between data distributions. In our framework, using a higher value of $\rho$ in random-data forgetting leads to more robust performance, as noted in Appendix C.2. Thank you for your detailed comment.
>
> ---
>
> **[Q5] Regarding the first-order Taylor approximation**
>
> Thank you for your comment. **We would like to emphasize that the linearity assumption is a technique commonly employed in generalization [7, 22] and adversarial robustness [10, 29, 37].** While this assumption may not hold in more complex models and datasets, these methods [7, 10, 22, 29, 37] have shown effective optimization in their respective domains. Specifically, sharpness-aware minimization [7, 22] demonstrates high performance when $\rho$ is in the range of $\rho \in$ {$0.005, 0.05, 0.5$}. Similarly, our method demonstrates stable performance across the range of $\rho$ values. The table below summarizes the results of class-wise forgetting on CIFAR-10.
>
> | **$\rho$** | **RA** | **FA** | **TA** | **$\Delta$Acc. $\downarrow$** | **MIA-Eff.** |
> | --- | --- | --- | --- | --- | --- |
> | **5e-3** | 99.99±0.00 (0.01) | 0.00±0.00 (0.00) | 94.32±0.66 (1.00) | 1.01 | 100.00±0.00 (0.00) |
> | **5e-2** | **100.0±0.00 (0.01)** | **0.00±0.00 (0.00)** | **94.51±0.63 (0.81)** | **0.82** | **100.00±0.00 (0.00)** |
> | **5e-1** | 100.0±0.00 (0.01) | 0.01±0.04 (0.01) | 93.33±1.08 (1.98) | 1.99 | 100.00±0.00 (0.00) |
> | **1e-1** | 99.65±0.31 (0.35) | 0.01±0.04 (0.01) | 92.66±1.12 (2.66) | 3.02 | 99.99±0.03 (0.01) |
>
> The table format is consistent with Table 1, except for the omission of time, as the results are based on the same computational setup. **Our method shows stable performance for a range of $\rho$, with $\Delta$Acc. values significantly better than those of previous unlearning frameworks (e.g., FT: 43.63, NG: 24.45, FF: 10.22, IU: 13.32).** We will definitely include an ablation section, including more results on various settings, in the modified version. Thank you again for your thoughtful comment.
>
> ---
>
> **[Q6-8] Typos and ambiguous sentences**
>
> We have corrected all typos and thoroughly reviewed the manuscript to address any other errors. Thank you for your careful reading.
>
> ---
>
> **Overall, we sincerely appreciate your detailed evaluation of our paper, and we are grateful for your comments, which have significantly improved the novelty of our work.**

---

> > ### Comment · Reviewer_QDnv · 2025-08-06
> > **Comment by Reviewer QDnv**
> >
> > Thank you for your responses, which have addressed my concerns.
> > I will adjust the score accordingly.

---

> ### Author Response · Authors · 2025-07-31
> **Rebuttal by Authors**
>
> **Supplementary References**
>
> [S1] Van Waerebeke, Martin, et al. "When to Forget? Complexity Trade-offs in Machine Unlearning." *Forty-second International Conference on Machine Learning*.
>
> [S2] Zhang, Binchi, et al. "Towards certified unlearning for deep neural networks." *Proceedings of the 41st International Conference on Machine Learning*. 2024.
>
> [S3] Aditya Golatkar, Alessandro Achille, and Stefano Soatto. “Eternal Sunshine of the Spotless
> Net: Selective Forgetting in Deep Networks”. In: IEEE/CVF Conference on Computer Vision
> and Pattern Recognition (CVPR). June 2020.
> [S4] Golatkar, Aditya, Alessandro Achille, and Stefano Soatto. "Forgetting outside the box: Scrubbing deep networks of information accessible from input-output observations." *European Conference on Computer Vision*. Cham: Springer International Publishing, 2020.
>
> [S5] Mu, Siqiao, and Diego Klabjan. "Rewind-to-delete: Certified machine unlearning for nonconvex functions." arXiv preprint arXiv:2409.09778 (2024).
>
> [S6] Chien, Eli, et al. "Certified machine unlearning via noisy stochastic gradient descent." Advances in Neural Information Processing Systems 37 (2024): 38852-38887.
>
> [S7] Machine unlearning: A survey Heng Xu, Tianqing Zhu, Lefeng Zhang, Wanlei Zhou, and Philip S Yu.
> ACM Computing Surveys, 56(1):1–36, 2023.
>
> ---
>
> **The following is a list of recent papers on LLM and vision unlearning from the past two years, published in top-tier conferences (71 papers in total).**
>
> For readability, author names have been omitted.
>
> - Generative Unlearning for Any Identity (CVPR 2024)
> - Not All Wrong is Bad: Using Adversarial Examples for Unlearning (ICML 2025)
> - A Certified Unlearning Approach without Access to Source Data (ICML 2025)
> - The WMDP Benchmark: Measuring and Reducing Malicious Use with Unlearning (ICML 2024)
> - PDUDT: Provable Decentralized Unlearning under Dynamic Topologies (ICML 2025)
> - SAeUron: Interpretable Concept Unlearning in Diffusion Models with Sparse Autoencoders (ICML 2025)
> - Verification of Machine Unlearning is Fragile (ICML 2024)
> - Unified Gradient-Based Machine Unlearning with Remain Geometry Enhancement (NeurIPS 2024)
> - Data Attribution for Text-to-Image Models by Unlearning Synthesized Images (NeurIPS 2024)
> - What Makes Unlearning Hard and What to Do About It (NeurIPS 2024)
> - Boosting Alignment for Post-Unlearning Text-to-Image Generative Models (NeurIPS 2024)
> - Towards Unbounded Machine Unlearning (NeurIPS 2023)
> - Fast Model Debias with Machine Unlearning (NeurIPS 2023)
> - Efficient Source-free Unlearning via Energy-Guided Data Synthesis and Discrimination-Aware Multitask Optimization (ICML 2025)
> - Targeted Unlearning with Single Layer Unlearning Gradient (ICML 2025)
> - Leveraging Per-Instance Privacy for Machine Unlearning (ICML 2025)
> - When to Forget? Complexity Trade-offs in Machine Unlearning (ICML 2025)
> - Certified Unlearning for Neural Networks (ICML 2025)
> - Knowledge Swapping via Learning and Unlearning (ICML 2025)
> - LoTUS: Large-Scale Machine Unlearning with a Taste of Uncertainty (CVPR 2025)
> - NoT: Federated Unlearning via Weight Negation (CVPR 2025)
> - Towards Source-Free Machine Unlearning (CVPR 2025)
> - MUNBa: Machine Unlearning via Nash Bargaining (ICCV 2025)
> - Unlearning the Noisy Correspondence Makes CLIP More Robust (ICCV 2025)
> - Robust Machine Unlearning for Quantized Neural Networks via Adaptive Gradient Reweighting with Similar Labels (ICCV 2025)
> - Meta-Unlearning on Diffusion Models: Preventing Relearning Unlearned Concepts (ICCV 2025)
> - Forgetting Through Transforming: Enabling Federated Unlearning via Class-Aware Representation Transformation (ICCV 2025)
> - Learning to Unlearn while Retaining: Combating Gradient Conflicts in Machine Unlearning (ICCV 2025)
> - Soft Prompting for Unlearning in Large Language Models (NAACL 2025)
> - Defensive Unlearning with Adversarial Training for Robust Concept Erasure in Diffusion Models (NeurIPS 2024)
> - Unveiling and Mitigating Backdoor Vulnerabilities based on Unlearning Weight Changes and Backdoor Activeness (NeurIPS 2024)
> - Shared Adversarial Unlearning: Backdoor Mitigation by Unlearning Shared Adversarial Examples (NeurIPS 2023)
> - Decoupled Distillation to Erase: A General Unlearning Method for Any Class-centric Tasks (CVPR 2025)
> - Unlearning through Knowledge Overwriting: Reversible Federated Unlearning via Selective Sparse Adapter (CVPR 2025)
> - Effective Skill Unlearning through Intervention and Abstention (NAACL 2025)
> - Model Sparsity Can Simplify Machine Unlearning (NeurIPS 2023)
> - SEMU: Singular Value Decomposition for Efficient Machine Unlearning (ICML 2025)
> - In-Context Learning (and Unlearning) of Length Biases (NAACL 2025)
> - LLM Unlearning via Embedding-Corrupted Prompts (NeurIPS 2024)
> - Ferrari: Federated Feature Unlearning via Optimizing Feature Sensitivity (NeurIPS 2024)
> - Fast Exact Unlearning for In-Context Learning Data for LLMs (ICML 2025)

---

> ### Author Response · Authors · 2025-07-31
> **Rebuttal by Authors**
>
> - Dissecting Fine-Tuning Unlearning in LLMs (Findings EMNLP 2024)
> - RWKU: Benchmarking Real-World Knowledge Unlearning for Large Language Models (ACL 2024)
> - Langevin Unlearning: A New Perspective of Noisy Gradient Descent for Machine Unlearning (NeurIPS 2024)
> - Certified Machine Unlearning via Noisy Stochastic Gradient Descent (NeurIPS 2024)
> - Certified Minimax Unlearning with Generalization Rates and Deletion Capacity (NeurIPS 2023)
> - NegMerge: Sign-Consensual Weight Merging for Machine Unlearning (ICML 2025)
> - Towards Certified Unlearning for Deep Neural Networks (ICML 2024)
> - Reminiscence Attack on Residuals: Exploiting Approximate Machine Unlearning for Privacy (ICCV 2025)
> - FG-OrIU: Towards Better Forgetting via Feature-Gradient Orthogonality for Incremental Unlearning (ICCV 2025)
> - Decoupling Memories, Muting Neurons: Towards Practical MU for LLMs (Findings ACL 2025)
> - Reversing the Forget–Retain Objectives: An Efficient LLM Unlearning Framework from Logit Difference (NeurIPS 2024)
> - From Evasion to Concealment: Stealthy Knowledge Unlearning for LLMs (Findings ACL 2025)
> - Large Language Model Unlearning (NeurIPS 2024)
> - Machine Unlearning of Pre-trained Large Language Models (ACL 2024)
> - Towards Safer LLMs through Machine Unlearning (Findings ACL 2024)
> - Unlearn What You Want to Forget: Efficient Unlearning for LLMs (EMNLP 2023)
> - Towards LLM Unlearning Resilient to Relearning Attacks: A Sharpness-Aware Minimization Perspective and Beyond (ICML 2025)
> - EFUF: Fine-Grained Unlearning Framework for Hallucination in Multimodal Large Language Models (EMNLP 2024)
> - SOUL: Unlocking the Power of Second-Order Optimization for LLM Unlearning (EMNLP 2023)
> - Revisiting Who’s Harry Potter: Targeted Unlearning via Causal Intervention Perspective (ACL 2024)
> - Fine-Grained Pluggable Gradient Ascent for Knowledge Unlearning in Language Models (EMNLP 2024)
> - CoME: An Unlearning-Based Approach to Conflict-Free Model Editing (NAACL 2025)
> - Unlearning as Multi-Task Optimization: A Normalized Gradient Difference Approach with an Adaptive Learning Rate (EMNLP 2024)
> - UNDIAL: Self-Distillation with Adjusted Logits for Robust Unlearning in Large Language Models (NAACL 2025)
> - Balancing Forget Quality and Model Utility: A Reverse KL-Divergence Knowledge Distillation Approach for Better Unlearning in LLMs (NAACL 2025)
> - Avoiding Copyright Infringement via Large Language Model Unlearning (NAACL 2025)
> - UNLEARN: Efficient Removal of Knowledge in Large Language Models (NAACL 2025)
> - GRU: Mitigating the Trade-off between Unlearning and Retention (ICML 2025)
> - Tool Unlearning for Tool-Augmented LLMs (ICML 2025)
> - Adaptive Localization of Knowledge Negation for Continual LLM Unlearning (ICML 2025 spotlight)

---

### Official Review · Reviewer_mZMi · 2025-07-02

**Clarity:** 3
**Significance:** 2
**Originality:** 2
**Rating:** 4
**Confidence:** 4

**Summary:**

This paper formulates machine unlearning as a minimax optimization problem. Specifically, it first maximizes the “forget” loss over a local delta-neighborhood of the parameters
w, and then minimizes the “retain” loss based on that adversarial perturbation. Drawing inspiration from Sharpness-Aware Minimization, the inner maximization admits a closed-form approximation (Theorem 1), and the authors further leverage a first-order Taylor expansion for computational efficiency.

**Questions:**

No more questions.

**Ethical Concerns:**

["NO or VERY MINOR ethics concerns only"]

**Final Justification:**

The authors have addressed all my concerns in their rebuttal and during the communication period. For this reason, I have raised the score from 2 to 4.

**Limitations:**

Yes

**Paper Formatting Concerns:**

No major formatting issue has been identified.

**Quality:**

2

**Strengths And Weaknesses:**

Strengths

The manuscript is well structured and clearly written, with a thorough and precise overview of related work.

Weaknesses

1. It omits discussion of a closely related adversarial training approach to unlearning [1]. Reference [1] also considers a minimax optimization formulation to address unlearning problems. Could the authors provide further explanations on how their approach differs from [1]?

2. The regularizer in Equation (2) compares losses at the optimal and approximate solutions but does not directly penalize the distance between them — why is that? Furthermore, the criterion for choosing $\epsilon$ to ensure that $w^*$ lies within the $\delta$-ball is unclear.

3. The experimental comparisons do not include several recent baselines [2–5], particularly in Tables 1 and 3. Moreover, there should be an ablation study on the perturbation range $\rho$.

4. The procedure for generating the optimization trajectories shown in Figure 1 is not clear. Which model and data were used to obtain these trajectories?

References

[1] Discriminative Adversarial Unlearning

[2] Large Language Model Unlearning via Embedding-Corrupted Prompts

[3] MUSE: Machine Unlearning Six-Way Evaluation for Language Models

[4] Negative Preference Optimization: From Catastrophic Collapse to Effective Unlearning

[5] Who's Harry Potter? Approximate Unlearning in LLMs

---

> ### Author Rebuttal · Authors · 2025-07-31
>
> We appreciate the reviewer’s evaluation of the strengths and weaknesses of our paper. Below, we provide a detailed response addressing each of the weaknesses [W#], with references to the main paper [#] as well as new supplementary references [S#].
>
> ---
>
> **[W1] Comparison to [S1]**
>
> Thank you for suggesting the reference [S1]. However, we believe the work in [S1] differs significantly from our approach. In [S1], adversarial unlearning is framed in the context of Generative Adversarial Networks (GANs), where two distinct networks (attacker and defender networks) are used. This differs fundamentally from our approach, which involves a min-max optimization procedure applied to a single network’s weight parameters, similar to sharpness-aware minimization [7, 22]. Moreover, adversarial robustness focuses on maximizing the input perturbation and minimizing an outer loss function, such as cross-entropy. In contrast, our work emphasizes optimization directly on the model parameters, which sets it apart from the traditional adversarial robustness framework.
>
> ---
>
> **[W2] Regarding penalizing distance to $w^*$ and clarity of algorithm**
>
> Equation (2) represents our theoretical analysis for integrating prior unlearning methods. In machine unlearning, the optimal solution **$w^*$** is assumed to be unknown due to the significant cost of obtaining it. Therefore, prior works [6, 22] focus on updating the model to remove the influence of forget data. In our work, rather than ignoring it, we replace the optimal solution with a surrogate weight based on the characteristic of **$w^*$**. Specifically, in equations (4) and (5), $\epsilon$ represents the concept that the surrogate weight should incur a higher forget loss than the current weight. Within the $\rho$-neighborhood in the weight space, there exists a weight with a higher loss, as has been previously discussed in other works [7, 22]. As evidence of the effectiveness of our approach, our method achieves a higher forget loss, as illustrated in Figures 1 and 2.
>
> For the clarity of our algorithm, we would like to emphasize that other reviewers, such as Reviewer MRvE, who stated "This paper presents a novel machine unlearning framework called Unlearning-Aware Minimization (UAM), which formulates unlearning as a min-max optimization problem," Reviewer QDnV, who mentioned "The key innovation of this paper lies in formulating the machine unlearning problem within a min-max optimization framework, which leads to promising empirical performance," and Reviewer NPr1, who remarked "Proposed methods are theoretically well described," all agreed with the contributions of our paper.
>
> ---
>
> **[W3] Regarding other baselines and ablation study on $\rho$**
>
> Thank you for suggesting the references [S2, S3, S4, S5]. While we appreciate these recommendations, **we believe that these works are not directly comparable** to ours for the following reasons:
>
> - **[S2]**: This work does not involve updating the model's parameters directly, which makes a direct comparison to our method unfair. Our approach specifically updates the weight distribution of the model, which is a key distinction.
> - **[S3]**: This is a benchmark dataset, not a baseline model, making it inappropriate for a direct comparison. We chose to use WMDP datasets in our work because they are widely used and facilitate easier comparison with other works in the field.
> - **[S4] and [S5]**: These approaches do not use a "retain set." In [S4], the focus is solely on the "forget set" during unlearning, whereas our method utilizes a general corpus as the "retain set." Similarly, [S5] proposes an unlearned model through interpolation between the target model and a reinforced model, but it does not involve a retain set. Therefore, a direct comparison with our approach is not feasible.
>
> Given these distinctions, we believe that the methods mentioned above are not as relevant to our work as RMU and others in Table 4, which align more closely with the goals and methods we propose. We would like to emphasize that **our goal is to present a unified framework that can be applied across various domains**, a feature that we consider a strength of our approach as **Reviewer MRvE noted, "UAM does not rely on specific loss functions or architectures, allowing it to be applied broadly across domains including image classification and LLM unlearning."** We will include a discussion of these works in the revised version of the paper.
>
> **For the ablation study,** we would like to note that an ablation study on Equation (8) with $\rho$ is already provided in Appendix B. However, we agree that including a sensitivity analysis on these hyperparameters under the main experimental settings would be valuable for the readers. The table below summarizes the results of class-wise forgetting on CIFAR-10.
>
> | **$\rho$** | **RA** | **FA** | **TA** | **$\Delta$Acc. $\downarrow$** | **MIA-Eff.** |
> | --- | --- | --- | --- | --- | --- |
> | **5e-3** | 99.99±0.00 (0.01) | 0.00±0.00 (0.00) | 94.32±0.66 (1.00) | 1.01 | 100.00±0.00 (0.00) |
> | **5e-2** | 100.0±0.00 (0.01) | 0.00±0.00 (0.00) | 94.51±0.63 (0.81) | 0.82 | 100.00±0.00 (0.00) |
> | **5e-1** | 100.0±0.00 (0.01) | 0.01±0.04 (0.01) | 93.33±1.08 (1.98) | 1.99 | 100.00±0.00 (0.00) |
> | **1e-1** | 99.65±0.31 (0.35) | 0.01±0.04 (0.01) | 92.66±1.12 (2.66) | 3.02 | 99.99±0.03 (0.01) |
>
> The experimental results for LLM unlearning are presented in the table below:
>
> | $\rho$ | MMLU $\uparrow$ | WMDP-Bio $\downarrow$ | WMDP-Cyber $\downarrow$ | $\Delta$Acc. $\downarrow$ |
> | --- | --- | --- | --- | --- |
> | **5e-2** | 0.3366(−0.2444) | 0.2710(−0.3660) | 0.2441(−0.1959) | -0.0755 |
> | **5e-3** | 0.5535(−0.0275) | 0.2655(−0.3715) | 0.2587(−0.1813) | -0.5253 |
> | **5e-4** | 0.5601(−0.0209) | 0.2727(−0.3643) | 0.2506(−0.1894) | -0.5328 |
> | **5e-5** | 0.5644(−0.0166) | 0.2930(−0.3440) | 0.2330(−0.2070) | -0.5344 |
>
> The table format is consistent with Table 4. As $\rho$ increases, harmful information decreases; however, this also leads to a reduction in MMLU performance. We observe that the best performance is achieved at $\rho = $ 5e-5. **We will definitely include an ablation section in the modified version to provide further insight into the contributions of different components of our method. Thank you again for your comment.**
>
> ---
>
> **[W4] Regarding Figure 1**
>
> In lines 189-191, we provide a detailed explanation of Figure 1. Specifically, we design a simple synthetic 2D optimization landscape with $w=[w_1,w_2]$, where the forget and retain losses are derived from rotated quadratic forms. The optimization is initialized at the point that minimizes the sum of the forget and retain losses, representing a pre-trained model. For further details, we refer the reader to Appendix C.1.
>
> ---
> **Overall, we sincerely appreciate your evaluation of our paper, and we are grateful for your comments. Thank you.**
>
> ---
>
> **Supplementary References**
>
> - [S1] Sharma, Rohan, et al. "Discriminative adversarial unlearning." arXiv preprint arXiv:2402.06864 (2024).
> - [S2] Liu, Chris, et al. "Large language model unlearning via embedding-corrupted prompts." *Advances in Neural Information Processing Systems* 37 (2024): 118198-118266.
> - [S3] Shi, Weijia, et al. "MUSE: Machine Unlearning Six-Way Evaluation for Language Models." *The Thirteenth International Conference on Learning Representations*.
> - [S4] Zhang, Ruiqi, et al. "Negative Preference Optimization: From Catastrophic Collapse to Effective Unlearning." *First Conference on Language Modeling*.
> - [S5] Eldan, R., and M. Russinovich. "Who’s Harry Potter? Approximate unlearning in LLMs, arXiv." *arXiv preprint arXiv:2310.02238* (2023).

---

> > ### Comment · Reviewer_mZMi · 2025-08-02
> > **responses to authors' rebuttal**
> >
> > I would like to thank the authors for the detailed rebuttal, which has addressed all my concerns. For this reason, I would like to raise my score to 4 (borderline accept).

---

> > > ### Author Response · Authors · 2025-08-02
> > >
> > > Thank you very much for your thoughtful response and for considering our rebuttal carefully. We truly appreciate your willingness to re-evaluate the paper and are grateful that our clarifications helped address your concerns. Again, thank you for your hard work.

---

### Official Review · Reviewer_NPr1 · 2025-07-03

**Clarity:** 2
**Significance:** 3
**Originality:** 3
**Rating:** 4
**Confidence:** 3

**Summary:**

This paper introduces a novel unlearning framework, Unlearning-Aware Minimization (UAM). UAM effectively removes forget set information while preserving retain set information by min-max optimization which involves inner maximization and outer minimization.

**Questions:**

- Although it showed superior unlearning performance compared to conventional methods, it seems not to work in a specific scenario where forget gradient is highly correlated with retain gradient. For example, in random data forgetting, if forget gradient from class 1 and retain gradient from class 1 may have same direction, then proposed objective function operates weirdly. Inferior unlearning performance in random data forgetting may be caused by this kind of reason. I want to ask the opinion of the authors about this concern.
- Typos: In caption of Figure 1, gradient ascent (GA) is used instead of negative gradient (NG). In Table 1 and Table 5, some values in blue are not matched with the actual difference from Retrain. Some values even disturb interpreting overall results.

**Ethical Concerns:**

["NO or VERY MINOR ethics concerns only"]

**Final Justification:**

The rebuttal addresses most concerns, however, it remains unclear why UAM is less effective than some comparison methods, such as $l1$-sparse, in random-data forgetting.

Therefore, the rating remains unchanged at borderline accept.

If the paper is accepted, it is recommended to revise the title "Unlearning-Aware Minimization", as it is unclear and does not accurately reflect the paper’s content.

**Limitations:**

- UAM introduces additional hyperparameters, such as γ and ρ, in addition to the learning rate. It may need efforts to search optimal hyperparameters for each task, dataset, and model.

**Quality:**

3

**Strengths And Weaknesses:**

### Strength:
- This paper proposes a novel unlearning method, Unlearning-Aware Minimization, which effectively remove forget data while maintaining general model performances.
- Proposed methods are theoretically well described.
- Proposed methods can be easily extended to other domains, such as multiple-choice question-answering using large language models.

### Weakness:
- Proposed methods introduce additional hyperparameters, such as γ and ρ, which needs to be tuned.
- In image classification, comparison is limited to the early unlearning methods. Comparison with recent advanced unlearning methods will strengthen the effectiveness of UAM.

---

> ### Author Rebuttal · Authors · 2025-07-30
>
> We appreciate the reviewer’s insightful evaluation of the strengths and weaknesses of our paper. We believe that your feedback provides valuable perspectives that will help enhance the quality and clarity of our work.
>
> Below, we provide a detailed response addressing each of the weaknesses [W#] and questions [Q#], with references to the main paper [#] as well as new supplementary references [S#].
>
> ---
>
> **[W1] Sensitivity analysis on $\rho$ and $\gamma$**
>
> Thank you for your valuable suggestion. First of all, we would like to note that an ablation study on Equation (8) with $\rho$ and Equation (11) with $\gamma$ is provided in Appendix B. However, we agree that including a sensitivity analysis on these hyperparameters under the main experimental settings would be valuable for the readers. The table below summarizes the results of the sensitivity experiment on $\rho$ and $\gamma$ for class-wise forgetting on CIFAR-10.
>
> | **$\rho$** | **RA** | **FA** | **TA** | **$\Delta$Acc. $\downarrow$** | **MIA-Eff.** |
> | --- | --- | --- | --- | --- | --- |
> | **5e-3** | 99.99±0.00 (0.01) | 0.00±0.00 (0.00) | 94.32±0.66 (1.00) | 1.01 | 100.00±0.00 (0.00) |
> | **5e-2** | 100.0±0.00 (0.01) | 0.00±0.00 (0.00) | 94.51±0.63 (0.81) | 0.82 | 100.00±0.00 (0.00) |
> | **5e-1** | 100.0±0.00 (0.01) | 0.01±0.04 (0.01) | 93.33±1.08 (1.98) | 1.99 | 100.00±0.00 (0.00) |
> | **1e-1** | 99.65±0.31 (0.35) | 0.01±0.04 (0.01) | 92.66±1.12 (2.66) | 3.02 | 99.99±0.03 (0.01) |
>
> | **$\gamma$** | **RA** | **FA** | **TA** | **$\Delta$Acc. $\downarrow$** | **MIA-Eff.** |
> | --- | --- | --- | --- | --- | --- |
> | **0** | 99.99±0.00 (0.00) | 63.59±6.13 (63.59) | 94.69±0.54 (0.63) | 64.22 | 95.80±2.66 (4.20) |
> | **1** | 99.99±0.01 (0.01) | 0.03±0.07 (0.03) | 94.57±0.63 (0.75) | 0.78 | 100.00±0.00 (0.00) |
> | **2** | 100.0±0.00 (0.01) | 0.00±0.00 (0.00) | 94.51±0.63 (0.81) | 0.82 | 100.00±0.00 (0.00) |
>
> The table format is consistent with that of Table 1, with the only difference being the omission of the time column, as these results are based on the same computational setup. For $\rho$, **our method demonstrates stable performance across a range of $\rho$ values, with $\Delta$Acc. significantly outperforming previous unlearning frameworks (e.g., FT: 43.63, NG: 24.45, FF: 10.22, IU: 13.32).** Regarding $\gamma$, as noted in Appendix B, using a positive value of $\gamma$ is crucial for achieving better $\Delta$Acc. Specifically, $\gamma=0$ implies that the Hessian information in Equation (8) is entirely ignored, which omits the core component of our method. **Using $\gamma = 2$ and $\rho = 0.05$ or $0.5$ results in stable performance across all domains and tasks.** The experimental results for LLM unlearning are presented in the table below:
>
> | $\rho$ | MMLU $\uparrow$ | WMDP-Bio $\downarrow$ | WMDP-Cyber $\downarrow$ | $\Delta$Acc. $\downarrow$ |
> | --- | --- | --- | --- | --- |
> | **5e-2** | 0.3366 (−0.2444) | 0.2710 (−0.3660) | 0.2441 (−0.1959) | -0.0755 |
> | **5e-3** | 0.5535 (−0.0275) | 0.2655 (−0.3715) | 0.2587 (−0.1813) | -0.5253 |
> | **5e-4** | 0.5601 (−0.0209) | 0.2727 (−0.3643) | 0.2506 (−0.1894) | -0.5328 |
> | **5e-5** | 0.5644 (−0.0166) | 0.2930 (−0.3440) | 0.2330 (−0.2070) | -0.5344 |
>
> The table format is consistent with Table 4. As $\rho$ increases, harmful information decreases; however, this also leads to a reduction in MMLU performance. We observe that the best performance is achieved at $\rho =$ 5e-5. Due to the character limit, we are unable to include the ablation on $\gamma$ on LLM unlearning.
> We will definitely include an ablation section in the modified version to provide further insight into the contributions of different components of our method. Thank you again for your thoughtful comment.
>
> ---
>
> **[W2] Validity of comparison methods**
>
> Thank you for your comment. We would like to clarify that while FT and NG are earlier methods, they remain strong baselines that are still widely used in recent work [4, S1, S2]. **In fact, concurrent research [S2, S3, S4] presented at ICML 2025 uses FT, NG, and l1-sparse as main comparison methods.** L1-sparse was presented at NeurIPS 2023, and SalUN, which we compared in Table 2, was introduced at ICLR 2024. Moreover, we argue that our framework can be integrated with recent methods [6, S3, S4, S5], as it is a fundamental training framework similar to FT and NG. For large language model (LLM) unlearning, we have also considered state-of-the-art methods such as RMU, which was presented at ICML 2024.
>
> ---
>
> **[Q1] Regarding the case when forget gradient and retain gradient are similar**
>
> Thank you for your insightful suggestions. Inspired by your idea, we first investigate the cosine similarity between the forget gradient, $\nabla L(w, D_f)$, and the retain gradient, $\nabla L(w, D_r)$, under random-data forgetting. To do this, with a fixed random seed (seed=42), we observe that the average cosine similarity is higher under random-data forgetting than under class-wise forgetting. Specifically, the cosine similarity in random-data forgetting is greater than that in class-wise forgetting. **However, it is noteworthy that the cosine similarity does not approach 1.** Instead, it fluctuates between -0.25 and 0.50 (unfortunately, due to reviewer guidelines, we are unable to include the figure here).
>
> Under the random-data forgetting setting, **we also observe that our method successfully disentangles the gradients.** The table below illustrates the range of cosine similarity across iterations during our unlearning:
>
> | Iteration | 0-500 | 500-1000 | 1000-1500 |
> | --- | --- | --- | --- |
> | Cosine value range | [-0.30,0.50] | [-0.25,0.25] | [0.00, 0.15] |
>
> Therefore, our method shows its effectiveness in identifying implicit features where the forget and retain gradients can be separated, leading to a decrease in the moving average of the cosine similarity. This suggests that our method is particularly useful when it is possible to distinguish between features to be forgotten and those to be retained.
>
> Now, let’s consider a thought experiment where $\nabla L(w, D_r)$ and $\nabla L(w, D_f)$ are identical for all iterations. This situation implies that the distributions of $D_r$ and $D_f$ are identical, making it impossible to disentangle these gradients. In this case, the forget accuracy should ideally not decrease, since any decrease would suggest different distributions. Interestingly, when the gradients are identical, the second term in (12) becomes constant. Only the loss $L(w, D_r)$ is primarily minimized. Therefore, even in this scenario, we argue that our method converges towards the optimal solution. This discussion can be verified by the results in Table 1, where our method shows stable performance across different random seeds, with a reduced accuracy gap. We appreciate your valuable comments and will include this discussion in the main text of the paper.
>
> ---
>
> **[Q2] Typos and blue numbers in Tables**
>
> Thank you for your comment. First, we would like to clarify that the blue values represent the average of the absolute differences between the "Retrain" method and each other method across different classes (or seeds). Therefore, the difference between the average values is not identical to the blue values. To explain this more clearly, let $v$ represent the value of each measure. Then, the following relationship holds: $\mathbb{E}[|v_{retrain}-v_{method}|] \neq \mathbb{E}[v_{retrain}] - \mathbb{E}[v_{method}].$ We understand that this could confuse the readers, and we will make sure to clarify this point in the final version of the paper.
>
> For typos, including GA, we have thoroughly reviewed the manuscript to address any other typographical errors. Thank you for your careful reading.
>
> ---
>
> **Overall, we truly appreciate your positive evaluation of our paper, and we are grateful for your detailed comments, which have significantly improved the novelty of our work.**
>
> ---
>
> **Supplementary References**
>
> - [S1] Huang, Zhehao, et al. "Unified gradient-based machine unlearning with remain geometry enhancement." Advances in Neural Information Processing Systems 37 (2024): 26377-26414.
> - [S2] Gu, Hanlin, et al. "Ferrari: federated feature unlearning via optimizing feature sensitivity." Advances in Neural Information Processing Systems 37 (2024): 24150-24180.
> - [S3] Ebrahimpour-Boroojeny, Ali, Hari Sundaram, and Varun Chandrasekaran. "Not All Wrong is Bad: Using Adversarial Examples for Unlearning." Forty-second International Conference on Machine Learning.
> - [S4] Cai, Zikui, Yaoteng Tan, and M. Salman Asif. "Targeted Unlearning with Single Layer Unlearning Gradient." Forty-second International Conference on Machine Learning.
> - [S5] Zhong, Zhengyi, et al. "Unlearning through knowledge overwriting: Reversible federated unlearning via selective sparse adapter." Proceedings of the Computer Vision and Pattern Recognition Conference. 2025.

---

> > ### Comment · Reviewer_NPr1 · 2025-08-03
> >
> > Thank you for the response. However, some concerns are not clearly addressed. Although the cosine similarity for random-data forgetting decreases with more iterations, UAM is still less effective than some comparison mehotds. Does the comparison method, especially $l1$-sparse, achieve a lower cosine similarity between forget and retain gradients? If so, it can undermine the significance of the proposed method. If not, they paper needs to clarify what factors actually affect the unlearning performance.

---

> ### Author Response · Authors · 2025-08-04
>
> Thank you for your insight. Based on your suggestion, we have conducted an additional experiment with L1-Sparse on CIFAR-10. **Our results show that L1-Sparse does not reduce the cosine similarity** between forget and retain gradients, unlike our method, as seen below:
>
> | Iteration        | 0-500  | 500-1000  | 1000-1500  |
> | -----------------| ------ | --------- | ---------- |
> | Cosine Value Range | [-0.2, 0.4] | [-0.2, 0.4] | [-0.2, 0.4] |
>
> We also evaluate the cosine similarity on the TinyImageNet experiment under random-data forgetting, where we believe you pointed out the lower effectiveness of our method. For our method, cosine similarity changes from [0, 0.25] to [0, 0.05], while L1-Sparse shows a change from [0, 0.3] to [0, 0.3]. Based on these observations, **we conclude that our gradient disentanglement mechanism is a unique characteristic of our method compared to L1-Sparse**.
>
> To elaborate on these results, **we suggest that there may be multiple approaches to achieving effective unlearning performance.** For example, in adversarial robustness, methods like AT [S1] and TRADES [S2] achieve robustness in different ways: AT minimizes loss on adversarial examples, while TRADES matches the logits of benign and adversarial examples. While they minimize different losses, they are still effective in achieving adversarial robustness. Similarly, our method presents a distinct approach to unlearning through min-max optimization, which geometrically encompasses the gradient disalignment effect, unlike regularization techniques such as L1-Sparse. SalUN, which we compare with our combined version in Table 2, can also be seen as a different approach compared to the previous ones, as it focuses on shifting only a limited set of parameters. **As highlighted by [20], machine unlearning, as it is still an emerging area of research, lacks a well-defined formal definition, making it difficult to establish standardized evaluation metrics. Therefore, we believe that further research is needed to develop new metrics and analyses for evaluating different approaches, which will remain an area for future work in this field.** We will discuss these results in the revised manuscript.
>
> **Thank you again for your valuable insight. Please don't hesitate to reach out if you have any further questions, and we truly appreciate it.**
>
> ---
> **Supplementary References**
>
> - [S1] Madry, Aleksander, et al. "Towards Deep Learning Models Resistant to Adversarial Attacks." International Conference on Learning Representations. 2018.
> - [S2] Zhang, Hongyang, et al. "Theoretically principled trade-off between robustness and accuracy." International conference on machine learning. PMLR, 2019.

---

### Official Review · Reviewer_MRvE · 2025-07-08

**Clarity:** 3
**Significance:** 3
**Originality:** 3
**Rating:** 4
**Confidence:** 3

**Summary:**

This paper presents a novel machine unlearning framework called Unlearning-Aware Minimization (UAM) , which formulates unlearning as a min-max optimization problem. It identifies model weights within a local neighborhood that maximize the loss on the data to be forgotten (forget data) and then minimizes the loss on the remaining data (retain data) using gradients from these surrogate weights. It effectively removes the influence of the forget data while maintaining performance on the retain data.

**Questions:**

See my detailed comments.

**Ethical Concerns:**

["NO or VERY MINOR ethics concerns only"]

**Final Justification:**

My all concerns are addressed. The additional experimental results show the effectiveness of the proposed method.

**Limitations:**

See my detailed comments.

**Paper Formatting Concerns:**

See my detailed comments.

**Quality:**

3

**Strengths And Weaknesses:**

**Strengths**:

Experimental results show that UAM achieves lower accuracy on the forget data (indicating successful unlearning) while preserving high accuracy on the retain data, surpassing the considered methods such as FT, NG

On LLM-based unlearning tasks, UAM significantly reduces the model's ability to generate harmful or hazardous content while retaining strong general performance (e.g., MMLU scores).

UAM does not rely on specific loss functions or architectures, allowing it to be applied broadly across domains including image classification and LLM unlearning..

**Weaknesses**:

1. The first-order Taylor approximation in Theorem 1 hinges on the assumption that "$\rho$ is sufficiently small." However, the paper lacks systematic validation of ρ’s impact across tasks. In high-dimensional or complex loss landscapes, this assumption may introduce non-negligible approximation errors, limiting the framework’s reliability.

2. The critique of Fine-Tuning (FT) and Negative Gradient (NG) is thorough, but the analysis of Fisher Forgetting (FF) and Influence Unlearning (IU) is superficial. The paper underestimates challenges unique to these methods, weakening the comparative evaluation

3. UAM is described as "scalable and efficient" but requires two gradient computations per iteration (inner/outer loops). Table 1 confirms the runtime of UAM exceeds FT and NG, contradicting efficiency claims and raising scalability concerns for larger models.

4. The abstract claims UAM "consistently outperforms existing methods," but in Table 3 (random forgetting), UAM exhibits higher $\Delta Acc.$ and worse MIA-Eff than $l_1$-sparse. This contradicts the universality of the outperformance claim.

5. In the experimental results comparison shown in Table 1, most of the compared methods are from 2022 or earlier, and there is a lack of comparisons with the latest state-of-the-art methods.

6. Proofs for Lemma 1 and Theorem 1 are relegated to the Appendix. Given their foundational role in deriving UAM, these should appear in the main text to facilitate immediate verification.

**Minor issues**:

1. Line 129: "sorely" should be "solely".
2. Line 140: "an weight point" should be "a weight point".

---

> ### Author Rebuttal · Authors · 2025-07-30
>
> We appreciate the reviewer’s insightful evaluation of the strengths and weaknesses of our paper. We believe that your feedback provides valuable perspectives that will help enhance the quality and clarity of our work.
>
> Below, we provide a detailed response addressing each of the weaknesses [W#] raised, with references to the main paper [#] as well as new supplementary references [S#].
>
> ---
>
> **[W1] Linearity assumption and systematic validation of ρ**
>
> Thank you for your comment. **We would like to emphasize that the linearity assumption is a technique commonly employed in generalization [7, 22] and adversarial robustness [10, 29, 37].** While this assumption may not hold in more complex models and datasets, these methods [7, 10, 22, 29, 37] have shown effective optimization in their respective domains. Specifically, sharpness-aware minimization [7, 22] demonstrates high performance when $\rho$ is in the range of $\rho \in$ {$0.005, 0.05, 0.5$}. Similarly, our method demonstrates stable performance across the range of $\rho$ values. The table below summarizes the results of class-wise forgetting on CIFAR-10.
>
> | **$\rho$** | **RA** | **FA** | **TA** | **$\Delta$Acc. $\downarrow$** | **MIA-Eff.** |
> | --- | --- | --- | --- | --- | --- |
> | **5e-3** | 99.99±0.00 (0.01) | 0.00±0.00 (0.00)  | 94.32±0.66 (1.00) | 1.01 | 100.00±0.00 (0.00) |
> | **5e-2** | **100.0±0.00 (0.01)** | **0.00±0.00 (0.00)**  | **94.51±0.63 (0.81)** | **0.82**  | **100.00±0.00 (0.00)** |
> | **5e-1** | 100.0±0.00 (0.01) | 0.01±0.04 (0.01)  | 93.33±1.08 (1.98) | 1.99 | 100.00±0.00 (0.00) |
> | **1e-1** | 99.65±0.31 (0.35) | 0.01±0.04 (0.01) | 92.66±1.12 (2.66) | 3.02 | 99.99±0.03 (0.01) |
>
> The table format is consistent with Table 1, except for the omission of time, as the results are based on the same computational setup. **Our method shows stable performance for a range of $\rho$, with $\Delta$Acc. values significantly better than those of previous unlearning frameworks (e.g., FT: 43.63, NG: 24.45, FF: 10.22, IU: 13.32).** We further believe that this topic could be explored more comprehensively by integrating our work with sharpness-aware minimization, specifically in terms of the success of linear approximation in these frameworks. The experimental results for LLM unlearning are presented in the table below:
>
> | $\rho$ | MMLU $\uparrow$ | WMDP-Bio $\downarrow$ | WMDP-Cyber $\downarrow$ | $\Delta$Acc. $\downarrow$ |
> | --- | --- | --- | --- | --- |
> | **5e-2** | 0.3366 (−0.2444) | 0.2710 (−0.3660) | 0.2441 (−0.1959) | -0.0755 |
> | **5e-3** | 0.5535 (−0.0275) | 0.2655 (−0.3715) | 0.2587 (−0.1813) | -0.5253 |
> | **5e-4** | 0.5601 (−0.0209) | 0.2727 (−0.3643) | 0.2506 (−0.1894) | -0.5328 |
> | **5e-5** | 0.5644 (−0.0166) | 0.2930 (−0.3440) | 0.2330 (−0.2070) | -0.5344 |
>
> The table format is consistent with Table 4. As $\rho$ increases, harmful information decreases; however, this also leads to a reduction in MMLU performance. We observe that the best performance is achieved at $\rho =$ 5e-5.
> We will definitely include an ablation section in the modified version to provide further insight into the contributions of different components of our method. Thank you again for your thoughtful comment.
>
> ---
>
> **[W2] Deeper analysis of FF and IU**
>
> First of all, both FT/NG and FF/IU are weak unlearning approaches; however, they differ in their unlearning algorithms. Specifically, FT and NG are retraining-based methods, requiring multiple epochs and training losses. **In contrast, FF and IU do not rely on epochs**; instead, they use Hessian-based information and perturb the weights only once. Our proposed framework aligns with retraining-based methods. **Indeed, several works in the retraining-based line of research [4, S1] do not include FF/IU, as these are not retraining-based methods.** However, similar to recent work [6], we believe it is important to at least include FF/IU in our comparison table to provide ease of comparison for researchers.
> We hope our method also contributes to the development of new unlearning frameworks based on Hessian-based algorithms such as FF/IU, which could relate to the insights presented in Lemma 2. We will incorporate this discussion into the Related Work section. Thank you for your feedback.
>
> ---
>
> **[W3] Regarding the "scalable and efficient" claim**
>
> Thank you for your comment. While it is true that our method requires two forward and backpropagation steps per iteration, compared to FT and NG, we would like to emphasize that FT and NG do not consistently achieve good unlearning performance, as demonstrated in Table 1. In this context, even though these methods may be computationally cheaper, they are not suitable for real-world applications due to their suboptimal unlearning performance. In contrast, our method demonstrates better unlearning performance.
>
> We would like to point out that a similar issue was raised with sharpness-aware minimization [7, 22], which also requires two forward and backward passes. Despite initial concerns regarding its computational cost, SAM has become a popular technique for improving performance. Subsequent research has led to more efficient versions [S2, S3], which have reduced the computational requirements. Similarly, we expect that faster techniques will emerge based on the principles outlined in our work. Regarding the claim of "scalability and efficiency," this specifically refers to the transition from Equation (10) to Equation (11). We will revise this section to ensure clarity in the modified version.
>
> ---
>
> **[W4] Regarding the "consistently outperforms existing methods" claim**
>
> We apologize for any confusion caused by the phrase "consistently outperforms existing methods." We will revise this claim in the modified version to better align with the actual findings and avoid overstating our results.
>
> ---
>
> **[W5] Validity of comparison methods**
>
> Thank you for your comment. We would like to clarify that while FT and NG are earlier methods, they remain strong baselines that are still widely used in recent work [4, S1, S4]. **In fact, concurrent research [S4, S5, S6] presented at ICML 2025 uses FT, NG, and L1-sparse as main comparison methods.** L1-sparse was presented at NeurIPS 2023. We would also like to emphasize that our framework is flexible and can be integrated with more recent approaches [6, S5, S6, S7], as it follows a foundational training framework similar to FT and NG. **A notable example of this is SalUN, introduced at ICLR 2024, which we compare with our combined version in Table 2.** For large language model (LLM) unlearning, we have also considered state-of-the-art methods such as RMU, which was presented at ICML 2024. We hope this clarifies our choice of comparisons and strengthens the rationale for the experimental design. Thank you for your valuable feedback.
>
> ---
>
> **[W6] Regarding the position of the proofs for Lemma 1 and Theorem 1**
>
> Thank you for your valuable suggestion. Due to page limitations, we moved the proofs for Lemma 1 and Theorem 1 to the Appendix. However, we acknowledge the importance of these proofs in the derivation of UAM. In the modified version, we will move these proofs to the main text, as we have an additional content page available. Thank you for your comment.
>
> ---
> **For minor issues,** we have corrected the typos in Line 129 ("sorely" to "solely") and Line 140 ("an weight point" to "a weight point"). We have also thoroughly reviewed the manuscript to address any other typographical errors. Thank you for your careful reading.
>
> ---
>
> **Overall, we sincerely appreciate your positive evaluation of our paper, and we are grateful for your comments, which have significantly improved the readability of our work.**
>
> ---
> **Supplementary References**
>
> - [S1] Huang, Zhehao, et al. "Unified gradient-based machine unlearning with remain geometry enhancement." Advances in Neural Information Processing Systems 37 (2024): 26377-26414.
> - [S2] Liu, Yong, et al. "Towards efficient and scalable sharpness-aware minimization." Proceedings of the IEEE/CVF Conference on Computer Vision and Pattern Recognition. 2022.
> - [S3] Mueller, Maximilian, et al. "Normalization layers are all that sharpness-aware minimization needs." Advances in Neural Information Processing Systems 36 (2023): 69228-69252.
> - [S4] Gu, Hanlin, et al. "Ferrari: federated feature unlearning via optimizing feature sensitivity." Advances in Neural Information Processing Systems 37 (2024): 24150-24180.
> - [S5] Ebrahimpour-Boroojeny, Ali, Hari Sundaram, and Varun Chandrasekaran. "Not All Wrong is Bad: Using Adversarial Examples for Unlearning." Forty-second International Conference on Machine Learning.
> - [S6] Cai, Zikui, Yaoteng Tan, and M. Salman Asif. "Targeted Unlearning with Single Layer Unlearning Gradient." Forty-second International Conference on Machine Learning.
> - [S7] Zhong, Zhengyi, et al. "Unlearning through knowledge overwriting: Reversible federated unlearning via selective sparse adapter." Proceedings of the Computer Vision and Pattern Recognition Conference. 2025.

---

> > ### Comment · Reviewer_MRvE · 2025-08-04
> > **Follow-up Comments**
> >
> > While I understand that FT, NG, and L1-sparse are commonly used baselines, the field has seen rapid progress recently. If other recent works also use these older methods for comparison, it doesn't justify overlooking newer state-of-the-art approaches. Given the growing number of advanced unlearning methods, especially in the past year, why haven't more recent and competitive methods been included in the evaluation?

---

> ### Author Response · Authors · 2025-08-05
>
> Thank you for reading our rebuttal and providing your feedback. Regarding the additional baselines, we would like to note that **we have encountered reproducibility issues with several papers during writing the paper.** Specifically, some papers either did not provide code or GitHub repositories, or the available repositories were not reproducible according to the results reported in the original papers. Therefore, as discussed in the main text of our paper, we have defined one of our key contributions as providing a more transparent repository, which we believe will make a positive contribution to the research community.
>
> **However, in response to your concerns, we revisited the most recent works presented at ICML 2025 [S1-S16].** From these, we selected the most reproducible methods based on their availability of algorithms and code, with minimal reliance on hyperparameters due to time and resource constraints during the rebuttal period. Given these rules, the two methods selected: SEMU [S1] and AMUN [S2]. Both of these focus on the random-data forgetting experiment, so we integrated them into our framework and conducted the additional experiments on CIFAR-10 under random-data forgetting. The results are summarized as follows:
>
> | Method | **RA** | **FA** | **TA** | $\Delta$Acc. ($\downarrow$) | **MIA-Eff.** |
> | --- | --- | --- | --- | --- | --- |
> | Retrain | 100.00±0.00 | 95.33±0.39 | 94.73±0.14 | 0.00 | 11.86±0.34 |
> | FT | 100.00±0.00 | 99.66±0.09 | 94.58±0.04 | 4.48 | 4.43±0.36 |
> | NG | 61.97±4.29 | 53.50±2.83 | 58.83±4.04 | 115.77 | 46.55±2.69 |
> | FF | 65.78±48.19 | 65.49±48.31 | 62.37±45.15 | 96.43 | 10.30±0.50 |
> | IU | 97.30±2.44 | 97.05±2.48 | 91.06±2.68 | 8.81 | 5.26±3.58 |
> | L1-sparse | 100.00±0.00 | 99.52±0.13 | 94.48±0.18  | 4.44 | 7.14±0.36 |
> | SEMU [S1] | 99.39±0.12 | 99.71±0.44 | 93.29±0.55 | 6.43 | 5.11±0.54 |
> | AMUN [S2] | 99.46±0.06 | 95.51±0.34 | 92.10±2.69 | 3.35 | 4.00±0.23 |
> | UAM | 99.88±0.03  | 95.19±0.28  | 92.74±0.29 | 2.53 | 9.16±0.14 |
>
> **As shown in the above table, our method shows the best performance with the lowest $\Delta$Acc.** Notably, SEMU, as reported in Table 3 of their original paper, does not show a reduction in forget accuracy compared to our method. Additionally, there is a larger gap between Retrain and our method in terms of MIA-Eff. For AMUN, we observed that while their forget accuracy is similar to ours, their test accuracy is lower, resulting in a higher $\Delta$Acc. compared to our approach. Given those results, we believe that these additional experiments support the effectiveness of our algorithm.
>
> **Thank you again for your thoughtful comment.** We remain committed to addressing these issues in future work, including further benchmarking and dataset development. **Please don't hesitate to reach out if you have any further questions, and we truly appreciate it.**
>
> ---
>
> **Supplementary References**
>
> - [S1] Not All Wrong is Bad: Using Adversarial Examples for Unlearning (ICML 2025)
> - [S2] SEMU: Singular Value Decomposition for Efficient Machine Unlearning (ICML 2025)
> - [S3] PDUDT: Provable Decentralized Unlearning under Dynamic Topologies (ICML 2025)
> - [S4] SAeUron: Interpretable Concept Unlearning in Diffusion Models with Sparse Autoencoders (ICML 2025)
> - [S5] Efficient Source-free Unlearning via Energy-Guided Data Synthesis and Discrimination-Aware Multitask Optimization (ICML 2025)
> - [S6] Targeted Unlearning with Single Layer Unlearning Gradient (ICML 2025)
> - [S7] Leveraging Per-Instance Privacy for Machine Unlearning (ICML 2025)
> - [S8] When to Forget? Complexity Trade-offs in Machine Unlearning (ICML 2025)
> - [S9] Certified Unlearning for Neural Networks (ICML 2025)
> - [S10] Knowledge Swapping via Learning and Unlearning (ICML 2025)
> - [S11] A Certified Unlearning Approach without Access to Source Data (ICML 2025)
> - [S12] NegMerge: Sign-Consensual Weight Merging for Machine Unlearning (ICML 2025)
> - [S13] Towards LLM Unlearning Resilient to Relearning Attacks: A Sharpness-Aware Minimization Perspective and Beyond (ICML 2025)
> - [S14] GRU: Mitigating the Trade-off between Unlearning and Retention (ICML 2025)
> - [S15] Tool Unlearning for Tool-Augmented LLMs (ICML 2025)
> - [S16] Adaptive Localization of Knowledge Negation for Continual LLM Unlearning (ICML 2025 spotlight)

---

> > ### Comment · Reviewer_MRvE · 2025-08-08
> > **Follow-up Comments**
> >
> > Thanks for the great efforts that authors have made. I hope the authors include these results in the revised paper. I would maintain my score.

---

> > > ### Author Response · Authors · 2025-08-08
> > >
> > > Thank you for your response. We are grateful that our clarifications helped address your concerns. Again, thank you for your hard work.

---

### Decision · Program_Chairs · 2025-09-17

**Decision:**

Accept (poster)

**Comment:**

**Summary:** This paper proposes Unlearning-Aware Minimization (UAM), a min-max optimization framework for machine unlearning that identifies weight perturbations maximizing forget loss, and then minimizes retain loss based on these surrogate weights. The method is inspired by sharpness-aware minimization and provides both a theoretical justification and efficient optimization. Experiments across image classification datasets and LLM-based unlearning tasks demonstrate that UAM outperforms strong baselines, achieving effective forgetting while preserving utility.

**Decision:** During the review and rebuttal process, several concerns were raised, including the reliance on first-order approximations, limited comparisons to the most recent baselines, scalability and efficiency claims, and weaker performance under random-data forgetting. The authors actively engaged with reviewers, adding ablations on hyperparameters, integrating newer baselines (e.g., SEMU, AMUN), and providing further analyses of gradient similarity and optimization behavior. Reviewers generally acknowledged these efforts, with some explicitly raising their scores after rebuttal. While certain limitations remain, the overall consensus is that the proposed framework makes a solid and timely contribution to the growing area of machine unlearning. I therefore recommend acceptance.